# Inhibitory mechanism of reveromycin A at the tRNA binding site of a class I synthetase

Bingyi Chen[1,2], Siting Luo[1,2], Songxuan Zhang[1,2], Yingchen Ju[1,2], Qiong Gu [2], Jun Xu [2], Xiang-Lei Yang [3] & Huihao Zhou[1,2 ✉]

The polyketide natural product reveromycin A (RM-A) exhibits antifungal, anticancer, anti-bone metastasis, anti-periodontitis and anti-osteoporosis activities by selectively inhibiting eukaryotic cytoplasmic isoleucyl-tRNA synthetase (IleRS). Herein, a co-crystal structure suggests that the RM-A molecule occupies the substrate tRNA[Ile] binding site of *Saccharomyces cerevisiae* IleRS (*Sc*IleRS), by partially mimicking the binding of tRNA[Ile]. RM-A binding is facilitated by the copurified intermediate product isoleucyl-adenylate (Ile-AMP). The binding assays confirm that RM-A competes with tRNA[Ile] while binding synergistically with L-isoleucine or intermediate analogue Ile-AMS to the aminoacylation pocket of *Sc*IleRS. This study highlights that the vast tRNA binding site of the Rossmann-fold catalytic domain of class I aminoacyl-tRNA synthetases could be targeted by a small molecule. This finding will inform future rational drug design.

[1] Guangdong Provincial Key Laboratory of Chiral Molecule and Drug Discovery, School of Pharmaceutical Sciences, Sun Yat-sen University, Guangzhou 510006, China. [2] Research Center for Drug Discovery, School of Pharmaceutical Sciences, Sun Yat-sen University, Guangzhou 510006, China. [3] Department of Molecular Medicine, Scripps Research Institute, La Jolla, CA 92037, USA. ✉email: zhuihao@mail.sysu.edu.cn

The polyketide reveromycin A (RM-A) (Fig. 1a) was originally isolated from *Streptomyces reveromyceticus* SN-593 in a study screening antitumour compounds in the early 1990s by the RIKEN Antibiotics Laboratory[1]. Reveromycin A could inhibit the proliferation of human tumour KB and K562 cells in vitro[2] and ovarian cancer and prostate cancer growth in vivo[3]. Later, RM-A was found to potently induce the apoptosis of osteoclasts (OCs) and suppress bone loss in ovariectomized mice[4]. RM-A is at least 100 times more cytotoxic to OCs than to a number of other types of cells, including OC progenitor cells, highlighting its potential in the development of anti-osteoporosis therapeutics[4,5]. The selectivity of RM-A to OCs is attributed to the three carboxyl moieties in its structure[4], making it highly hydrophilic and difficult to penetrate the cell membrane. To resorb bone, mature OCs form an acidic microenvironment, which suppresses proton dissociation of RM-A and increases its cell permeability. For the same reason, RM-A also inhibits bone metastasis[6] and has antiperiodontitis[7] activity. In addition, under acidic conditions, RM-A exhibits high toxicity to other cells[4] and fungi[2,8], such as *Candida albicans*, with an MIC value of 3 μM at pH 3.0[2]. Using a genetic approach in *Saccharomyces cerevisiae*, the molecular target of RM-A was identified as cytoplasmic isoleucyl-tRNA synthetase (IleRS)[9].

IleRS belongs to the aminoacyl-tRNA synthetase (AARS) family. Living cells usually contain a set of AARSs consisting of twenty members, and each AARS corresponds to a specific proteinogenic amino acid. AARSs catalyse the attachment of amino acids to their cognate tRNAs to form "charged" tRNAs, which will be subsequently escorted to ribosomes to translate genetic codes[10,11]. As indispensable components of the protein translation apparatus, inhibition of any AARS could impair the cellular protein translation process and thus suppress cell growth. Therefore, AARSs are attractive targets for developing drugs to treat infectious[12] and noninfectious diseases such as cancer and fibrotic diseases[13]. Two AARS inhibitors, mupirocin[14] and AN2690[15], are currently used in the clinic. Importantly, while traditional inhibitors mostly target the amino acid and/or ATP binding pockets of AARSs, new mechanistic inhibitors targeting other pockets of AARSs are highly desired. RM-A consists of a 6,6-spiroketal core with a hemisuccinate side chain, two unsaturated side chains with a terminal carboxylic acid moiety, and two alkyl side chains (Fig. 1a). This structure of RM-A does not resemble any substrate or other known inhibitor of AARSs. Moreover, RM-A is highly specific to eukaryotic IleRSs (IC$_{50}$ is approximately 2-10 nM for yeast and human IleRSs)[16–18], and it does not show significant inhibition against bacterial IleRSs[2,5] or

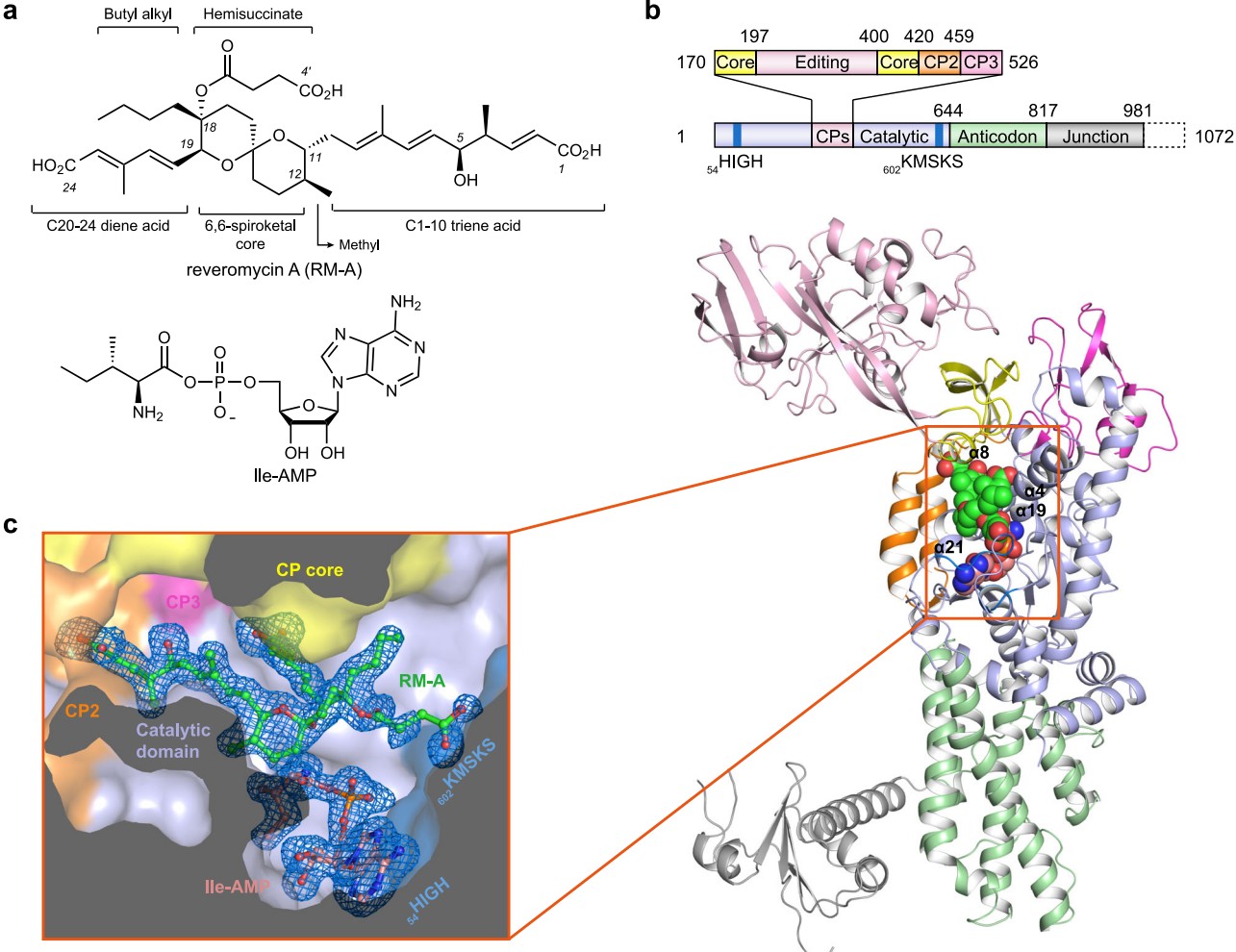

**Fig. 1 Structure of *Sc*IleRS with bound ligands. a** Chemical structures of reveromycin A (RM-A) and the intermediate product Ile-AMP. **b** Overview of the *Sc*IleRS • RM-A • Ile-AMP complex structure determined at a resolution of 1.9 Å. The residue numbers in the colour-coded diagram indicate domain boundaries. RM-A and Ile-AMP are shown in spherical models. The colour schemes for RM-A (green) and Ile-AMP (salmon) are the same throughout the manuscript. The α-helices forming the binding pocket of RM-A are labelled. **c** An annealed omit electron density map of RM-A and intermediate Ile-AMP calculated with Fourier coefficients F$_o$ - F$_c$ and contoured at 2.5 σ. RM-A and Ile-AMP co-bind in the aminoacylation pocket of the catalytic domain.

other AARS family members[9]. While its potent and selective biological activities make RM-A valuable for drug development, the molecular mechanism underlying the specific inhibition of eukaryotic IleRS by RM-A remains unknown.

Here, we show a co-crystal structure of *S. cerevisiae* IleRS (*Sc*IleRS) in complex with RM-A at 1.9 Å resolution. The structure shows that RM-A mainly occupies the binding site of the 3' CCA end of tRNA$^{Ile}$ and partially conflicts with the phosphate groups of ATP. Using the three carboxylic acid side chains, RM-A mimics the key interactions between IleRS and its natural substrates. Surprisingly, a molecule of the intermediate product isoleucyl-adenylate (Ile-AMP) copurified with *Sc*IleRS binds together with RM-A to the active site. Biophysical assays indicate that both an Ile-AMP analogue and the substrate L-isoleucine could enhance the affinity of RM-A to IleRS. The polyketide natural product RM-A reveals an amino acid/intermediate-aided AARS inhibitory mechanism and highlights the potential of using the vast tRNA binding site of class I AARSs for drug discovery in the future.

## Results

**Overview of the *Sc*IleRS · RM-A · Ile-AMP ternary complex.** To elucidate the eukaryotic IleRS-specific inhibitory mechanism of RM-A, a C-terminal truncated form of *S. cerevisiae* cytoplasmic IleRS (residues 1-984) was co-crystallized with RM-A, and the complex structure was determined to a resolution of 1.9 Å with $R/R_{free}$ factors of 17.7%/19.4% (Supplementary Table 1 and Supplementary Fig. 1). As a class Ia AARS, the overall structure of

*Sc*IleRS consists of a Rossmann-fold catalytic domain (residues 8-169, 527-643), connective polypeptide (CP) core (residues 170-196, 400-419), CP1 editing domain (residues 197-399), CP2 (residues 420-458), CP3 (residues 459-526), anticodon-binding domain (residues 644-816), and C-terminal junction domain (residues 817-981) (Fig. 1b). The extensive electronic densities in the active site cavity of the catalytic domain can fit simultaneously with two distinct ligands: an intermediate product Ile-AMP and an RM-A (Fig. 1c). The binding pocket of RM-A is mainly constituted by three helices (α4, α19 and α21) and two class I AARS signature motifs (HIGH and KMSKS) from the catalytic domain and the sequence linking CP2 and CP3 subdomains (linker CP2-3, $_{453}$ARDWNVSRNRYWG$_{465}$). In addition, helix α8 of the CP core forms a lid partially covering the binding pocket of RM-A (Fig. 1b, c).

**Structural basis of the recognition of RM-A by *Sc*IleRS.** The 6,6-spiroketal core of RM-A is located in the centre of the pocket, and the side chains of RM-A stretch in multiple directions to interact extensively with *Sc*IleRS. The C1-10 triene acid segment, the largest side chain of RM-A with a hydrophobic arm and an acidic end, lies on the crevice between helices α19 and α21 from the catalytic domain and helix α8 from the CP core (Fig. 2a). Residues Trp449, Trp456, Trp529 and Tyr571 form hydrophobic interactions to stabilize the triene arm. Arg454 forms ionic interactions with the terminal carboxyl group (C1), and Arg460 and Asp527 form H-bonds with the C5 hydroxyl group. In addition, the structural water molecules Wat34 and Wat36

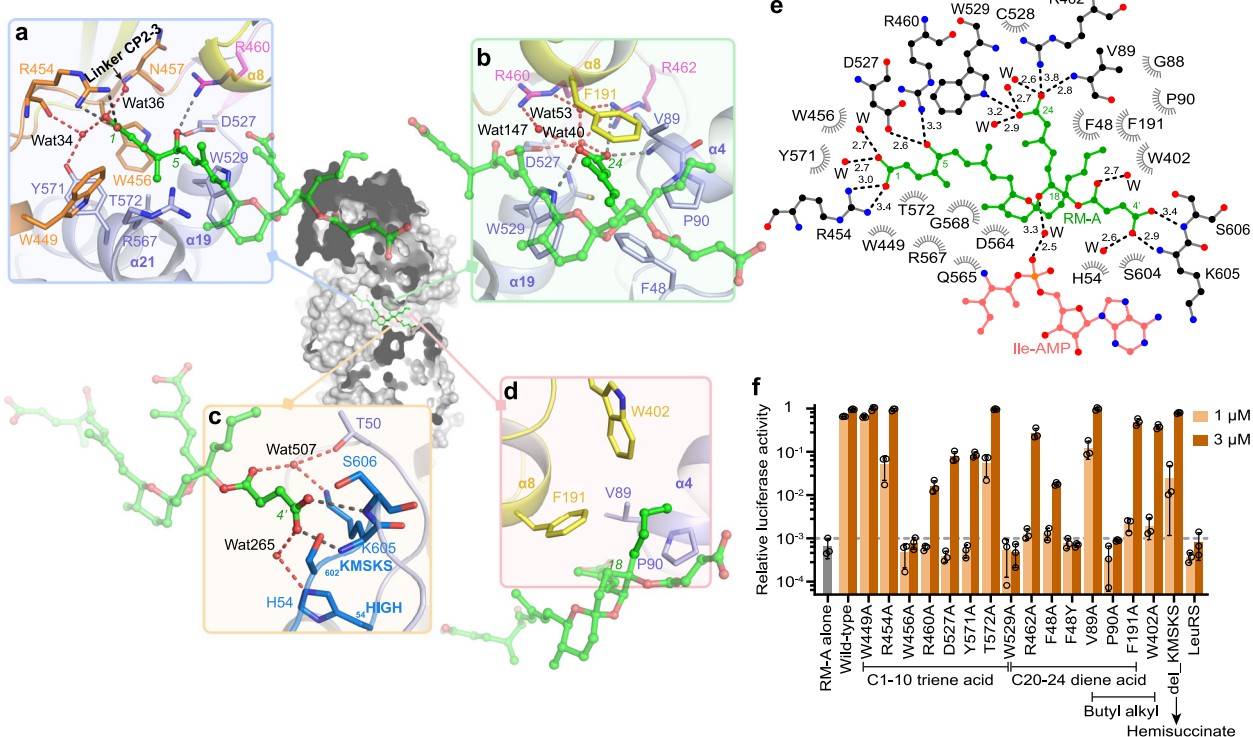

**Fig. 2 RM-A interacts with *Sc*IleRS from multiple directions and forms extensive polar interactions and hydrophobic contacts with *Sc*IleRS. a** Zoomed-in view of the binding site of the C1-10 triene acid segment of RM-A. The hydrogen bonds formed with water molecules (red spheres) are indicated as red dashed lines, and the polar interactions between RM-A and the residues of *Sc*IleRS are described as black dashed lines. **b–d** Zoomed-in view of the binding sites of the C20-24 diene acid moiety (**b**), the hemisuccinate (**c**), and the butyl alkyl side chain (**d**) of RM-A. **e** Two-dimensional presentation of RM-A binding. The ligands RM-A and Ile-AMP and the residues forming polar interactions with RM-A are shown in ball-and-stick representations, and other residues within 4.5 Å of RM-A are shown in grey. The H-bonding structural water molecules (W) are shown in red spherical models. **f** C-terminal truncated *Sc*IleRS or its variants at 1 μM or 3 μM were added to rabbit reticulocyte lysate to test their capability to rescue the protein translation inhibited by 200 nM RM-A. Almost all of the mutations, except W449A, strongly reduced the rescue capability of *Sc*IleRS. All error bars represent standard deviations (SD) of three independent (*n* = 3) experiments, and the data are presented as mean values ± SD. Source data are provided as a Source Data file.

contribute H-bonding interactions with the C1 carboxyl group (Fig. 2a, e). Structure-activity relationship (SAR) studies of RM-A analogues showed that derivatives with esterified C1 carboxyl group (2) or without C1-4 portion (5) only weakly inhibit the aminoacylation activity of IleRS and that further shortening of the C1-10 side chain (6, 7) will result in lower inhibition activity (Supplementary Fig. 2)[16,17], highlighting the important roles of these extensive hydrophobic and polar interactions between ScIleRS and the C1-10 triene acid segment for RM-A binding.

The C20-24 diene acid moiety lies between helices α4 and α19 from the catalytic domain and helix α8 from the CP core and is roughly parallel to the C1-10 triene acid (Fig. 2b). Phe48, Val89, Pro90, Phe191 and Trp529 form hydrophobic interactions with the diene chain. The C24 carboxyl group is deeply buried in the pocket and forms ionic interactions with Arg462 as well as multiple H-bonding interactions with residues Trp529 and Val89 and structural waters Wat40, 53 and 147 (Fig. 2b, e). Consistently, the methyl esterification of the C24 carboxyl group (3), which impairs the polar interactions between the C24 carboxyl group and IleRS, abolished the inhibitory effect of RM-A against IleRS activity and protein synthesis in vitro, while the methyl esterification of the C1 or C4' carboxyl group (2, 4) still maintained weak inhibitory activity against IleRS (70 or 90 times less active than RM-A, respectively) (Supplementary Fig. 2)[16].

The hemisuccinate at C18 of RM-A points in a direction opposite to the above two side chains, and its C4' carboxyl group forms two H-bonds with the backbones of the KMSKS loop. In addition, water molecules mediate the indirect interactions between hemisuccinate and Thr50 and His54 (HIGH motif) (Fig. 2c, e).

RM-A also contains two alkyl side chains on the 6,6-spiroketal core. The smaller methyl chain on C12 is close to helix α21. The butyl chain at C18 is roughly perpendicular to the three side chains with terminal carboxyl groups and stretches outwards from the aminoacylation pocket. This butyl chain forms hydrophobic interactions with Val89, Pro90, Phe191 and Trp402 (Fig. 2d, e). Spirofungin A (SF-A), an analogue of RM-A that only bears a small methyl group at C18, could only weakly inhibit IleRS (Supplementary Fig. 2)[17].

To further recognize the inhibitory mode of RM-A to ScIleRS, we then introduced mutagenesis analysis into the residues interacting with RM-A. RM-A potently inhibited the enzyme activity of ScIleRS as well as the translation of luciferase mRNA in the rabbit reticulocyte lysate (Supplementary Fig. 3). The truncated ScIleRS (residues 1-984) could bind with RM-A but did not charge tRNA^Ile (Supplementary Fig. 4a) due to the lack of a C-terminal tRNA-interacting domain[14]. Thus, the addition of C-terminal truncated ScIleRS to the rabbit reticulocyte lysate neutralized RM-A and rescued the protein translation inhibited by RM-A (Fig. 2f). As a control, the addition of leucyl-tRNA synthetase (LeuRS) did not show any rescue effect. Our data showed that all of the ScIleRS variants, except W449A, which only contributes to weak hydrophobic interactions, largely lost their rescue capability. In particular, the variants F48Y, P90A, W456A and W529A completely lost rescue capability even at a concentration of 3 μM (Fig. 2f). We also measured the binding of RM-A to full-length ScIleRS variants by employing a fluorescence-based thermal shift assay (TSA) and isothermal titration calorimetry (ITC) (Supplementary Table 2 and Supplementary Figs. 5 and 6). These data further highlighted the importance of Trp456, Arg460, Tyr571, Trp529, Arg462, Phe48, and Pro90 for RM-A binding because the mutations on those residues reduced the affinity of RM-A to ScIleRS by more than 10-fold. These mutagenesis analyses supported the structural observations regarding the binding mode of RM-A in the active site of ScIleRS.

The sequence alignments revealed that the residues involved in RM-A binding are highly conserved among eukaryotic cytoplasmic IleRSs (Supplementary Fig. 7). Molecular docking suggested that RM-A would inhibit human IleRS, the cellular target of RM-A in the treatment of osteoporosis and human cancers, with the same mechanism used to inhibit ScIleRS (Supplementary Fig. 8a). Consistently, when the residues corresponding to the RM-A-binding residues of ScIleRS were mutated, the C-terminal truncated human IleRS largely lost the ability to rescue the in vitro translation inhibited by RM-A (Supplementary Fig. 8b).

**RM-A cooperates with Ile or Ile-AMP for ScIleRS binding.** An Ile-AMP molecule was unambiguously modelled in the active site cavity of ScIleRS according to the 1.9 Å resolution density map, although none of the ATP, Ile, or Ile-AMP analogues were added during either protein purification or crystallization. RM-A lies approximately 4–5 Å above Ile-AMP. A water molecule bridges an oxygen of the 6,6-spiroketal core of RM-A to the α-phosphate group of Ile-AMP (Figs. 2e and 3a). Moreover, the isoleucyl group of Ile-AMP, the residues Phe48 and Trp529 from the catalytic domain, the hydrophobic portions of RM-A, and the residue Phe191 from the CP core form a four-layer hydrophobic sandwich (Fig. 3a). Mutation of these three residues, particularly Phe48 and Trp529, to Ala dramatically reduced the RM-A binding of ScIleRS (Fig. 2f, Supplementary Table 2 and Supplementary Figs. 5 and 6), supporting the importance of hydrophobic stacking in RM-A inhibition.

We employed TSA to study the binding of RM-A and other ligands to ScIleRS. The $T_m$ value of full-length ScIleRS was approximately 54 °C. The presence of ATP, L-isoleucine, Ile-AMS (a nonhydrolysable analogue of Ile-AMP) and RM-A increased the $T_m$ values of ScIleRS by approximately 4 °C, 8 °C, 23 °C, and 7 °C, respectively (Fig. 3b). RM-A in combination with ATP, L-isoleucine or Ile-AMS increased the $T_m$ values of ScIleRS by approximately 7 °C, 13 °C, or 29 °C, respectively. These results indicated that RM-A could bind to full-length ScIleRS cooperatively with L-isoleucine or Ile-AMS but not with ATP. The $T_m$ value of C-terminal truncated ScIleRS was not improved by ATP and L-isoleucine (Supplementary Fig. 4b), consistent with that of ScIleRS bound with the copurified intermediate Ile-AMP. When an additional Ile-AMP analogue (Ile-AMS) was supplemented to achieve a higher concentration of Ile-AMP/Ile-AMS, a co-shift between Ile-AMP/Ile-AMS and RM-A was then observed (Supplementary Fig. 4b), supporting the co-binding of Ile-AMP/Ile-AMS and RM-A to the C-terminal truncated ScIleRS as well. In contrast, RM-A could bind to neither the apo Staphylococcus aureus IleRS (SaIleRS) nor SaIleRS saturated with L-isoleucine or Ile-AMS (Fig. 3c).

In addition, we measured the isotherms of titrating RM-A to ScIleRS. The disassociation constant ($K_d$) for RM-A binding to apo ScIleRS was determined to be $164 \pm 10$ nM, and it was improved to $95 \pm 12$ nM for ScIleRS saturated with Ile-AMS (Fig. 3d, e). These results supported the co-binding of Ile-AMP and RM-A to ScIleRS. Surprisingly, the substrate L-isoleucine was found to enhance the binding of RM-A to IleRS even more effectively, as the $K_d$ of RM-A to ScIleRS saturated with L-isoleucine was determined to be $17 \pm 2$ nM (Fig. 3f).

Structural analysis of the class I AARSs methionyl-tRNA synthetase (MetRS) and LeuRS showed that the binding of the substrate amino acids could rearrange the amino acid binding sites, resulting in the aromatic residues (corresponding to Phe48 and Trp529 of ScIleRS) flipping to cover the hydrophobic side chain of the substrate amino acids[19–21] (Supplementary Fig. 9). In our crystal structure of ScIleRS bound with RM-A and Ile-AMP,

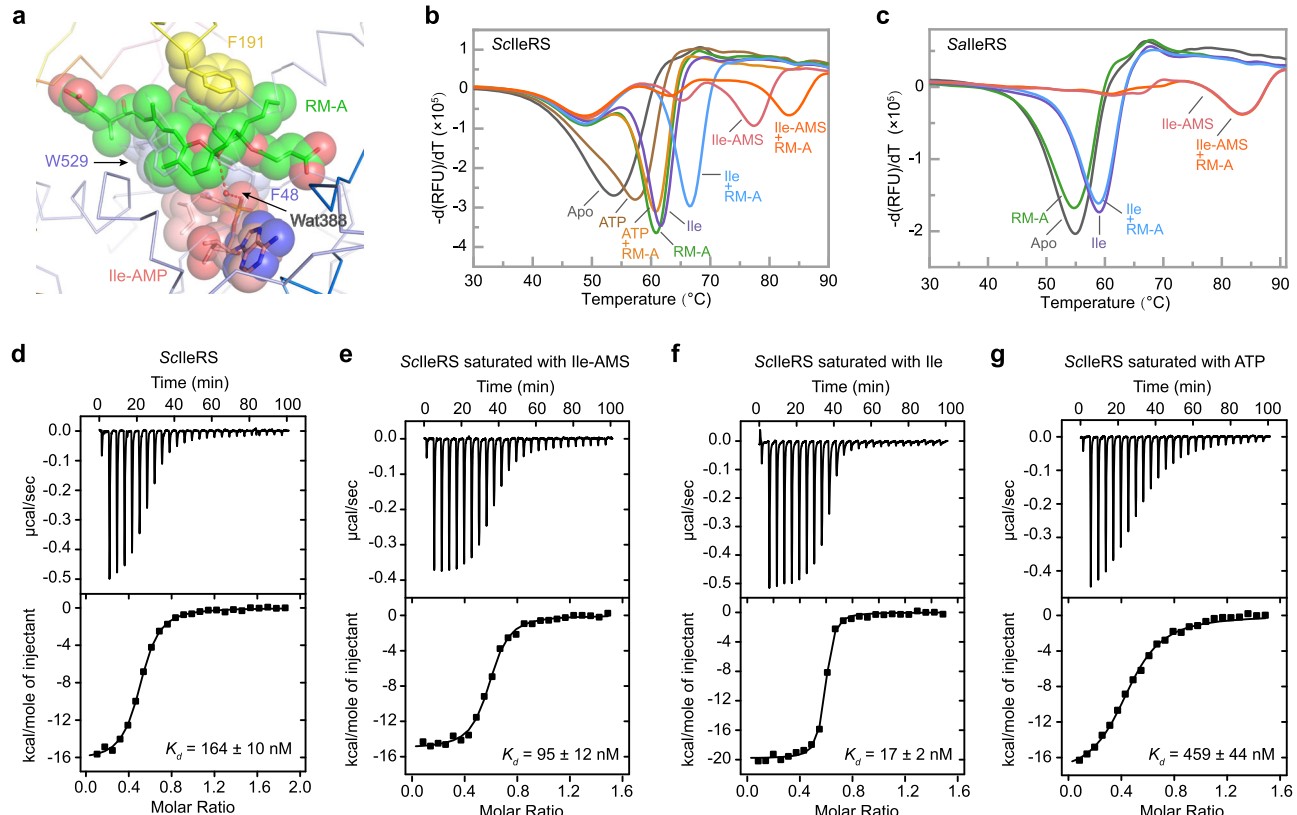

**Fig. 3 The cooperative binding of RM-A and l-isoleucine or Ile-AMP. a** Close-up view of the interactions between RM-A and Ile-AMP. Ile-AMP and RM-A and residues F48, F191 and W529, which form hydrophobic stacking interactions, are displaced as spherical models. The water-mediated H-bonding interactions are shown as red dashed lines. **b** Thermal melting curves of ScIleRS in the presence of different ligands. RM-A alone could improve the $T_m$ of ScIleRS by approximately 7 °C. Moreover, RM-A could co-bind with L-isoleucine and Ile-AMS but not ATP. **c** Thermal shift curves of SaIleRS in the presence of different ligands revealed that RM-A could not bind to SaIleRS, even in the presence of L-isoleucine or the intermediate analogue Ile-AMS. **d–g** ITC titrations of RM-A to apo ScIleRS (**d**) or ScIleRS protein presaturated with Ile-AMS (**e**), L-isoleucine (**f**) or ATP (**g**). The binding isotherms were fitted to a single-site model.

Phe48 adopted a conformation similar to its corresponding residue Tyr15 in *E. coli* MetRS bound with L-methionine and Tyr43 in *E. coli* LeuRS bound with L-leucine, and it was sandwiched by the isoleucyl group of Ile-AMP and the C20-C24 diene acid moiety of RM-A to form hydrophobic stacking (Fig. 3a). The binding of L-isoleucine or the isoleucyl group of Ile-AMP to IleRS may facilitate the rearrangement of Phe48 and then form the important hydrophobic interaction network, which revealed an amino acid/intermediate-assisted binding mechanism of an inhibitor to AARS. However, why L-isoleucine enhanced the binding affinity of RM-A more efficiently than Ile-AMP is still unclear.

**RM-A occupies the binding sites of ATP phosphate groups.** The KMSKS loop is a class I AARS signature motif that contributes to binding the phosphate groups of substrate ATP and stabilizing the transition state during aminoacylation[22], and it was observed to form two H-bonds with the C4′ carboxyl group of RM-A in our structure of the ScIleRS · RM-A · Ile-AMP complex (Fig. 2c). Because the ScIleRS · ATP complex structure is not currently available, we compared our structure with other class I AARS · ATP structures to analyse the possible competitive feature of RM-A with the substrate ATP. When the structures of the tryptophanyl-tRNA synthetase (TrpRS) · ATP complex (PDB ID: 1M83)[23] and tyrosyl-tRNA synthetase (TyrRS) · ATP complex (PDB ID: 1H3E)[24] were superimposed on ScIleRS · RM-A · Ile-AMP, the adenosine moieties within the ATP and Ile-AMP molecules were well aligned, whereas the β- and γ-phosphates of

ATP might clash with the C18 hemisuccinate of RM-A (Fig. 4a). When ScIleRS was superimposed on MetRS (PDB ID: 3KFL)[25], the C18 hemisuccinate also clashed with the unreleased pyrophosphate (Fig. 4b). These analyses suggested that RM-A uses its C4′ carboxyl group to mimic the interactions of ATP phosphate groups with the KMSKS loop and might prevent ATP from binding to the active site of IleRS. Consistently, the TSA assay revealed that RM-A and ATP could not bind to IleRS cooperatively (Fig. 3b). We also measured the isotherm of titrating RM-A to ScIleRS premixed with ATP, and the $K_d$ of 459 ± 44 nM (Fig. 3g) was approximately 2.8-fold larger than the $K_d$ (164 ± 10 nM, Fig. 3d) of titrating apo protein, indicating the existence of competition between ATP and RM-A in binding to ScIleRS.

To our surprise, when $_{600}$GRKMSKSLK$_{608}$ was replaced by a short linker (-GSGS-), the C-terminal truncated ScIleRS variant (3 μM) could still rescue approximately 80% of the protein translation inhibited by RM-A (Fig. 2f). In contrast, some of the single mutations on C1-10 triene acid and C20-24 diene acid-binding residues completely abolished the rescue capability of the C-terminal truncated IleRS (Fig. 2f). Consistently, the affinity of RM-A to the corresponding variant of full-length ScIleRS only slightly decreased compared with wild-type ScIleRS (Supplementary Fig. 6 and Supplementary Table 2). These results suggested that the interactions between the C4' carboxyl group and the KMSKS loop are less important for the binding of RM-A to ScIleRS. Interestingly, reveromycin T (RM-T), the precursor of RM-A in biosynthesis that lacks hemisuccinate (Supplementary Fig. 2), showed stronger inhibitory activity against IleRS than RM-A[18],

suggesting that competition with ATP is not necessary for the inhibitory activity of RM-A and its analogues. It is possible that without the hemisuccinate group, RM-T is able to co-bind to IleRS with Ile+ATP as well as Ile-AMP, and the broadened specificity may contribute to the improved inhibitory activity of RM-T.

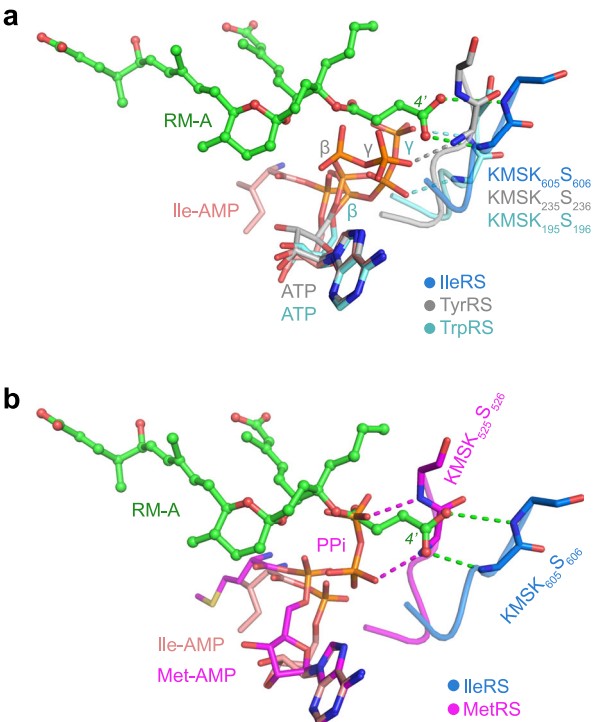

**Fig. 4 The C18 hemisuccinate of RM-A uses its terminal carboxyl group to mimic the phosphate groups of ATP for IleRS binding. a** Structural superposition of the *T. thermophilus* TyrRS • ATP complex (grey, PDB ID: 1H3E) and *Bacillus stearothermophilus* TrpRS • ATP complex (cyan, PDB ID: 1M83) with the *Sc*IleRS • RM-A • Ile-AMP complex (blue). The backbones of the KMSKS loop form H-bonds with the C4' carboxyl group of RM-A (indicated as green dashed lines), and they also form similar H-bonds with the β- or γ-phosphates of ATP in ATP-bound TyrRS and TrpRS structures (indicated as grey and cyan dashed lines, respectively). **b** Structural superposition of the *Leishmania major* MetRS • Met-AMP • PPi ternary complex (magenta, PDB ID: 3KFL) with the *Sc*IleRS • RM-A • Ile-AMP complex. The H-bonds between the backbones of the KMKSK loop and PPi are indicated as magenta dashed lines.

### RM-A represents the first tRNA-site inhibitor of class I AARSs.

RM-A is located above Ile-AMP in the active site cavity. Similarly, in class I AARSs, the CCA end of substrate tRNA also binds here to accept the aminoacyl moiety from the intermediate aminoacyl-adenylate. The structure of LeuRS, one of the closest homologues of IleRS in the AARS family, in complex with tRNA$^{Leu}$ at the aminoacylation state has been determined (PDB ID: 4AQ7)[21]. Superimposition of the catalytic domain of *Sc*IleRS to that of LeuRS found substantial overlaps between RM-A and the tRNA CCA end in synthetase binding (Fig. 5a).

The 6,6-spiroketal core and the diene acid side chain of RM-A largely overlap with the stacked bases C75 and A76 of tRNA$^{Leu}$, and the butyl chain at C18 creates a clash with the ribose of C75 (Fig. 5b). The C24 carboxyl group of RM-A is located at a similar site as the backbone phosphate group of A76. While the phosphate group of A76 forms an H-bond with Leu84 and ionic interactions with Arg426 in *Ec*LeuRS • tRNA complex[21], the C24 carboxyl group of RM-A forms similar interactions with the corresponding residues Val89 and Arg462 of *Sc*IleRS (Fig. 5b). In addition, while Arg424 of *Ec*LeuRS interacts with the phosphate group of C74 of tRNA$^{Leu}$, Arg460 of *Sc*IleRS forms an H-bond with the C5 hydroxyl group of RM-A (Fig. 5b). Although the C1-10 triene acid chain of RM-A lies in the pocket at a higher position than the U72A73C74 backbone of tRNA$^{Leu}$, the C1-6 portion of RM-A stretches to the binding site of A73. The C1 carboxyl group of RM-A forms ionic interactions with Arg454 of *Sc*IleRS, which is similar to the H-bonding interaction between Arg418 of *Ec*LeuRS and the 2'-OH of the A73 ribose of tRNA$^{Leu}$ (Fig. 5b). Thus, the structural comparison between the *Sc*IleRS • RM-A • Ile-AMP complex and LeuRS • tRNA suggests that by partially mimicking the binding mode of substrate tRNA, RM-A occupies the binding pocket of the tRNA 3' CCA end in the catalytic domain of IleRS, which prevents the productive binding of tRNA for aminoacylation.

To probe the competition, we measured the $K_d$ of RM-A to *Sc*IleRS premixed with in vitro transcribed tRNA$^{Ile}$. Indeed, the binding affinity decreased approximately 5-fold (to a $K_d$ of 763 ± 178 nM, Fig. 5c) compared with binding to the apo protein (Fig. 3d). As a control, tRNA$^{Pro}$ did not disturb the binding

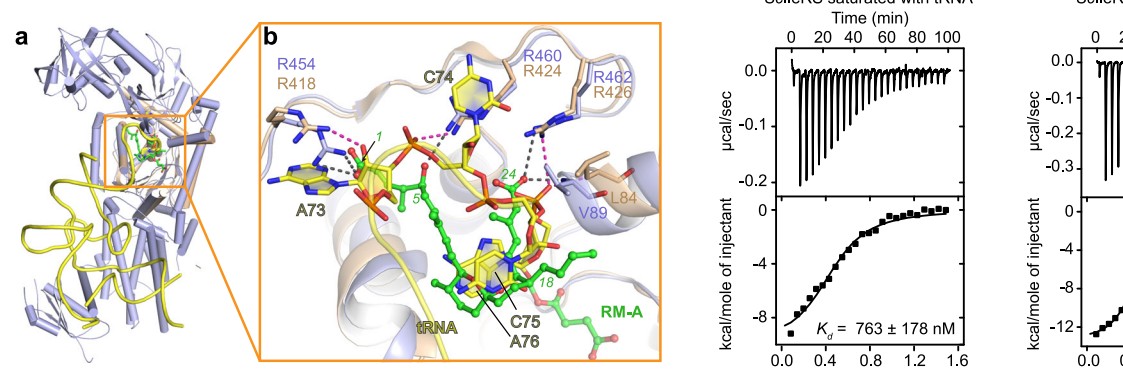

**Fig. 5 RM-A represents the first tRNA-site inhibitor of class I AARSs. a** Structural superimposition of the catalytic domain of the *E. coli* LeuRS • tRNA$^{Leu}$ complex (wheat, PDB ID: 4AQ7) with that of the *Sc*IleRS • RM-A • Ile-AMP complex (light blue) reveals that a large part of RM-A occupies the binding site of the tRNA 3' CCA end. **b** Zoomed-in view of the conflicts between RM-A and the 3' CCA end of tRNA. The residues that form polar interactions (indicated as black dashed lines) with the C1-10 and C20-24 side chains of RM-A in *Sc*IleRS are shown in sticks, and the corresponding residues in *E. coli* LeuRS were found to form H-bonds (indicated as magenta dashed lines) with A74, C75 and A76 of tRNA$^{Leu}$. **c, d** ITC titrations of RM-A to *Sc*IleRS premixed with in vitro transcribed tRNA$^{Ile}$ (**c**) and with tRNA$^{Pro}$ (**d**).

between RM-A and *Sc*IleRS ($K_d = 275 \pm 17$ nM, Fig. 5d). Thus, RM-A can compete with tRNA$^{\text{Ile}}$ for IleRS binding.

**Structural explanation for the insensitivity of bacterial IleRS and other AARSs to RM-A.** Similar to other AARS members, bacterial and eukaryotic IleRSs exhibit significant similarity in terms of sequence, structure, and the mechanisms of substrate recognition and catalysis. RM-A potently inhibits the activity of eukaryotic IleRSs; however, it was reported to be inactive against bacterial IleRSs[2]. Consistently, our TSA data showed that RM-A did not bind to bacterial IleRS, even in the presence of L-isoleucine or Ile-AMS (Fig. 3c).

When looking at the residues within 4.5 Å of RM-A, the hydrophobic interactions between the C1-10 triene acid segment of RM-A and residues Phe191, Trp449, Tyr571 and Thr572 of *Sc*IleRS were largely lost in bacterial IleRSs because of the nonconserved residue substitutions (Supplementary Figs. 7 and 10). These substitutions in bacterial IleRSs may prevent the binding of the long triene acid segment of RM-A because the *Sc*IleRS variants F191A, Y571A and T572A were shown to be less capable of binding RM-A (Fig. 2f, Supplementary Table 2 and Supplementary Figs. 5 and 6).

Sequence analysis also revealed that Phe48 between RM-A and the isoleucyl portion of Ile-AMP was conserved in eukaryotic IleRSs and was replaced by Tyr in bacterial IleRSs (Supplementary Figs. 7 and 10). Interestingly, the F48Y variant of C-terminal truncated *Sc*IleRS failed to rescue translation even at 3 μM (Fig. 2f). Consistently, TSA results showed that RM-A, both alone and together with Ile or Ile-AMS, bound to the F48Y variant of full-length *Sc*IleRS weakly, and further ITC results showed that the affinity of RM-A to the F48Y variant was 17-fold lower than that to wild-type *Sc*IleRS (Supplementary Table 2 and Supplementary Figs. 5 and 6). Most likely, the additional hydroxyl group of Tyr compared to Phe will reduce the hydrophobic stacking effect in the four-layer hydrophobic sandwich (Fig. 3a).

In addition, helix α4 in *Sc*IleRS forms a strong H-bond with the most important carboxyl group (C24) of RM-A, but the interaction could be lost in *Sa*IleRS because its α4 shifts away by approximately 2.2 Å (Supplementary Fig. 10). This structural difference might be due to a short insertion (helix α5) in eukaryotic IleRSs that does not exist in bacterial IleRSs (Supplementary Figs. 7 and 10). In general, the absence of a large number of interactions may be responsible for the insensitivity of bacterial IleRSs to RM-A.

LeuRS and valyl-tRNA synthetase (ValRS), the closest homologues of IleRS in class I AARS, were also reported to be insensitive to RM-A[9]. We modelled RM-A to LeuRS and ValRS by overlaying the structure of the catalytic domain of the *Sc*IleRS · RM-A · Ile-AMP complex to that of human LeuRS (PDB ID: 6LPF)[26] and *T. thermophilus* ValRS (PDB ID: 1GAX)[27], and many RM-A binding residues of *Sc*IleRS were found to be substituted in these two AARSs (Supplementary Figs. 11a, b and 12). For example, the key hydrophobic stacking residue Phe48 of *Sc*IleRS changes to Tyr in LeuRS and to Asn in ValRS, and in addition, the other key hydrophobic residue Trp529 changes to Leu in LeuRS, which could weaken the hydrophobic stacking interactions important for RM-A binding. Furthermore, the polar interactions contributed by Arg454 and Arg462 of *Sc*IleRS are also partially abolished in eukaryotic LeuRS and ValRS. Moreover, some residues on α21 of *Sc*IleRS are substituted with larger residues in LeuRS and ValRS, which possibly cause clashes with the C1-10 triene acid segment of RM-A, and mutating Thr571 to either Tyr (the corresponding residue in LeuRS) or Arg (the corresponding residue in ValRS) blocked the binding of RM-A to *Sc*IleRS, as shown by TSA (Supplementary Fig. 11c).

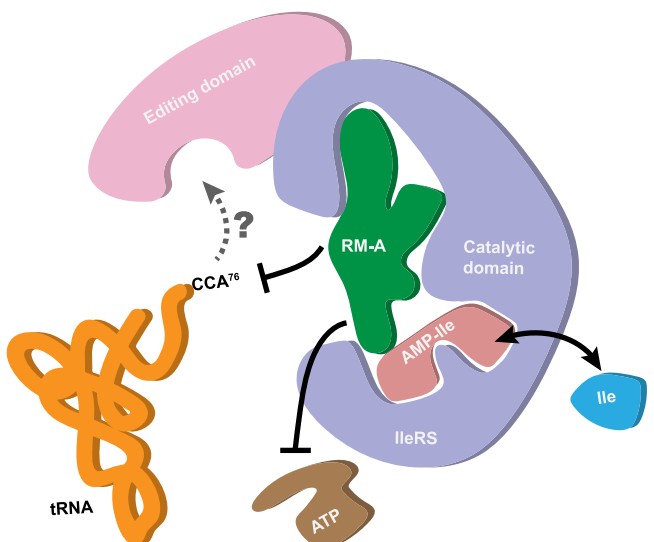

**Fig. 6 Schematic diagram of the inhibition of eukaryotic IleRS by RM-A.** RM-A occupies the binding site of the 3′ CCA end of tRNA$^{\text{Ile}}$ on the catalytic domain of eukaryotic cytoplasmic IleRS. Thus, it blocks the productive binding of tRNA$^{\text{Ile}}$ to the catalytic domain, while the alternative binding of tRNA$^{\text{Ile}}$ to the editing domain of IleRS might not be affected. RM-A may also partially conflict with the phosphate groups of ATP. In contrast, the substrate L-isoleucine and the intermediate product Ile-AMP could co-bind with RM-A to the aminoacylation pocket of IleRS.

## Discussion

Since its isolation in the early 1990s, antibiotic RM-A has been extensively investigated for its promising bioactivities in treating multiple human diseases[1–7]. Many RM-A analogues have been isolated as natural products or chemically synthesized, resulting in the discovery of more effective analogues as well as providing SAR understandings of this class of molecules[8,16–18]. Here, after thirty years of investigation of its biological activities, our co-crystal structure showed that RM-A binds to *Sc*IleRS at its pocket for the CCA end of the cognate tRNA in the presence of the copurified intermediate Ile-AMP. RM-A reveals a substrate/intermediate-aided tRNA site-inhibitory mechanism (Fig. 6), which is unique to all known small molecule inhibitors of AARS enzymes. Moreover, the cooperative binding between RM-A and amino acid/aminoacyl-adenylate highlights the potential of developing drug combinations to more effectively target AARSs.

The binding sites for ATP, amino acids and the tRNA CCA end, all of which are necessary for aminoacylation, can be exploited to develop AARS inhibitors. Most of the currently found aminoacylation inhibitors target the ATP and/or amino acid binding sites of AARSs[28,29]. For example, the well-studied mupirocin and the nonhydrolysable intermediate analogues can simultaneously occupy the ATP and amino acid binding sites to achieve high inhibition potency[14,29]. The antibiotic cladosporin[30] against lysyl-tRNA synthetase (LysRS) and the natural product resveratrol[31] against tyrosyl-tRNA synthetase (TyrRS) are representative single-site inhibitors targeting the ATP and amino acid binding sites, respectively. However, to date, no single-site inhibitor targeting the tRNA 3′ CCA end binding site has been found. One possible reason is that nucleotide A76 at the CCA end of tRNA is structurally similar to ATP, so that a molecule that mimics A76 may be preferentially captured by the ATP binding site, which provides stronger interactions[28,32]. The concentration of ATP can reach 1–10 mM in cells[33], so ATP competitive inhibitors need both high potency and high cellular concentration to be effective. Therefore, many ATP competitive inhibitors of

AARSs exhibit significantly lower potency in cell-based systems than in vitro with purified enzymes. Recently, the herb-derived medicine halofuginone (HF) was found to competitively bind to both proline and tRNA A76 binding sites of eukaryotic prolyl-tRNA synthetase (ProRS) in an ATP-dependent manner[34], resulting in promising antifibrosis and anticancer activities. The tRNA-site inhibitory mechanism of HF was successfully applied to develop other class II AARS inhibitors[35,36]. Moreover, the natural product borrelidin (BN)[37] was identified to inhibit threonyl-tRNA synthetase (ThrRS) by simultaneously occupying the binding sites of all three substrates, including the CCA end of tRNA[Thr]. However, both ProRS and ThrRS belong to class II AARS, and no molecules had been identified to inhibit the tRNA site of class I AARSs until the present study of RM-A.

The tRNA acceptor arm adopts a hairpin structure covering the aminoacyl-adenylate in the vast active site cavity of the Rossmann-fold catalytic domain of class I AARSs[21]. The hairpin structure of the CCA end was stabilized by extensive protein-tRNA and intra-tRNA interactions; thus, chemicals mimicking the structure of A76 or the linear tRNA CCA end are unlikely to bind to class I AARSs with high affinity. By comparing with the structure of LeuRS bound with tRNA[Leu], it is suggested that rather than resembling the structure of substrate tRNA, RM-A uses a special scaffold to partially mimic the interactions between tRNA and IleRS. Notably, although RM-A also forms H-bonds with the KMSKS loop of IleRS by using its C18 hemisuccinate chain, these interactions do not seem necessary for the binding of RM-A to IleRS. RM-T, the analogue of RM-A without C18 hemisuccinate, showed even higher inhibitory activity against IleRS than RM-A[18]. Moreover, it is possible that the ester bond is unstable in vivo, and the hydrolysis product without hemi-succinate may play an important role in the in vivo activity of RM-A. Thus, we propose that RM-A and its analogues are predominantly tRNA-site inhibitors rather than tRNA-ATP dual-site inhibitors, and this study on RM-A revealed that extensive interactions with the CCA end binding site alone are sufficient to endue inhibitors with strong biological activity.

In conclusion, the bioactive natural product RM-A was revealed in this study to use a unique stereo scaffold to potently inhibit the tRNA CCA end binding site of eukaryotic IleRS, and the substrate L-isoleucine or intermediate Ile-AMP may play an important role in facilitating inhibition. Because class I AARSs all use the conserved Rossmann-fold catalytic domain to catalyse the charging of tRNAs at the CCA end[38], the scaffold and inhibitory mechanism of RM-A are, in principle, applicable for the development of tRNA-site inhibitors for other class I AARSs. These inhibitors can be used in combination with molecules targeting other binding sites in the same enzyme and/or with each other to target multiple AARSs to overcome potential drug resistance.

## Methods

**Protein preparation.** The DNA fragment encoding the full-length cytoplasmic ScIleRS (residues 1-1072) was amplified from the genomic DNA of S. cerevisiae (ATGG 204508) and inserted into the pET20b plasmid (Novagen) using Nde I and Xho I restriction sites, with a hexahistidine tag at its C-terminus. The transformed E. coli BL21 (DE3) Condon Plus cells were grown in Luria-Bertani (LB) medium supplemented with 100 mg/L ampicillin at 37 °C until the $OD_{600}$ reached ~0.6, and then 0.2 mM isopropyl-β-D-thiogalactoside (IPTG) was added to induce protein overexpression at 20 °C for 20 h. The E. coli cells were harvested by centrifugation, and the cell pellet was resuspended and sonicated in washing buffer (500 mM NaCl; 20 mM Tris-HCl, pH 8.0; 5% v/v glycerol; 5 mM β-mercaptoethanol; and 10 mM imidazole). The cell lysate was centrifuged at 4000 × g for 30 min, and the supernatant was then loaded onto a Ni-NTA column (Qiagen) pre-equilibrated with washing buffer. The impurity was washed away with 20 column volumes of washing buffer, and then the target protein was eluted with 5 column volumes of elution buffer (500 mM NaCl; 20 mM Tris-HCl, pH 8.0; 5% v/v glycerol; 5 mM β-ME; and 100 mM imidazole). The protein was further purified by HiLoad 16/60 Superdex 200 pg (GE healthcare) in running buffer (20 mM HEPES, pH 7.5;

150 mM NaCl; 5 mM β-ME; and 5% v/v glycerol). The full-length ScIleRS was concentrated to 10 mg/mL and stored at −80 °C before use.

The DNA sequence encoding the C-terminal truncated ScIleRS (residues 1–984) or C-terminal truncated human IleRS (residues 1–983) was subcloned into pET29b (Novagen) with a C-terminal hexahistidine tag. The expression and purification of the C-terminal truncated proteins are similar to those of the full-length ScIleRS protein. The C-terminal truncated ScIleRS used for crystallization was desalted and concentrated to 90 mg/mL and stored at −80 °C in buffer containing 50 mM NaCl, 5 mM Tris-HCl (pH 8.0), and 5 mM β-ME. The ScIleRS or human IleRS variants were expressed and purified using the same strategy as wild-type proteins, and the sharp and symmetric peaks in gel filtrations indicated that they were well folded. All primers used to construct recombinant plasmids are listed in Supplementary Table 3.

**Crystallography.** The sitting-drop vapour-diffusion method was employed to crystallize the ScIleRS · RM-A complex. The C-terminal truncated ScIleRS (residues 1-984) (50 mg/mL) was preincubated with 2 mM RM-A (Nova Chemistry) at 4 °C for 20 min. Each drop consisted of 1 μL of protein solution and 1 μL of reservoir solution (0.2 M di-ammonium tartrate, 20% PEG 3350) and was equilibrated against 100 μL of reservoir solution at 8 °C for 3–7 days to allow crystals to grow. Large crystals were immersed in a cryoprotectant solution (0.2 M di-ammonium tartrate, 20% PEG3350, and 20% ethylene glycol) for a few seconds and then flash-frozen in liquid nitrogen. The diffraction data were collected using a single crystal at 100 K with a wavelength of 0.979 Å at beamline BL17U1 of the Shanghai Synchrotron Radiation Facility (SSRF) and were integrated and scaled using HKL2000[39]. The structure was solved by molecular replacement using the T. thermophilus IleRS structure (PDB ID: 1ILE)[40] as the search model in the program Molrep[41]. Iterative refinements of the structure model were carried out using Coot[42] and Refmac5[43]. The stereochemical quality of the final model was assessed using MolProbity[44]. The statistics of the data collection and structural refinement are listed in Supplementary Table 1. The coordinate and structural factors of the ScIleRS · RM-A · Ile-AMP complex have been deposited in Protein Data Bank (PDB) under the accession code 7D5C.

**Fluorescence-based thermal shift assay.** The binding of RM-A to IleRS and the impact of intermediate analogue (Ile-AMS, MedChemExpress) or substrates on binding were evaluated by the fluorescence-based thermal shift assay (TSA). Ligand binding usually stabilizes the protein during the thermal denaturation process, and tighter binding usually causes a larger positive shift in the protein melting temperature ($T_m$)[45]. Briefly, mixtures with a final volume of 20 μL containing 150 mM NaCl, 100 mM HEPES (pH 7.5), 10% glycerol, 5 mM DTT, 2 μg of IleRS, 4× SYPRO orange fluorescence dye (Sigma-Aldrich) and different ligands (50 μM RM-A, 2 mM L-isoleucine, 2 mM ATP, or 50 μM Ile-AMS) were prepared in 96-well plates. The mixtures were incubated at 25 °C for 10 min and then heated from 25 °C to 95 °C at a rate of 1 °C/min. The fluorescence intensity was recorded by a StepOnePlus Real-Time PCR instrument (Life Technologies), and the maximums of the first derivatives were determined to calculate $T_m$ values. Each $T_m$ value was an average of triple assays.

**In vitro translation assay.** The inhibitory effect of RM-A against protein translation was measured in a rabbit reticulocyte lysate (Promega). RM-A (from 3 μM to 1.5 nM) was added to the 1.5-fold diluted rabbit reticulocyte lysate supplemented with 0.02 mg/mL firefly luciferase mRNA (Promega). After incubating at 30 °C for 1.5 h, 2.5 μL of the reaction was transferred to a 384-well plate and mixed with 25 μL of Luciferase Assay Reagent (Promega). The luminescence was read on a FlexStation 3 multimode microplate reader (Molecular Devices). The luciferase activity of the reaction without adding RM-A was normalized to 1. The $IC_{50}$ value of RM-A against in vitro protein translation was calculated by fitting the curve of relative luciferase activity versus RM-A concentration using the dose-response (inhibition) function in GraphPad Prism 7 (GraphPad Software). The same in vitro translation assays were employed to test the capability of different proteins in rescuing the protein translation suppressed by RM-A. The lysates were mixed with 200 nM RM-A, 1 μM or 3 μM C-terminal truncated IleRSs or their variants or E. coli LeuRS and incubated for 1.5 h at 30 °C. All rescue experiments were performed in triplicate, and the error bars are SD.

**In vitro transcription of tRNA.** The DNA templates with a T7 promoter upstream of the sequence encoding tRNA[Ile](GAU) or tRNA[Pro](UGG) were constructed by two consecutive PCRs[46]. The primers are listed in Supplementary Table 4. To increase transcription by T7 RNA polymerase, the wild-type base-pair A1-U72 of E. coli tRNA[Ile](GAU) was changed to the G1-C72 pair[47], and the nucleotide at the 5′ end (C1) of E. coli tRNA[Pro](UGG) was deleted[48]. The transcription reactions were performed at 37 °C for 5 h in a mixture containing 200 mM Tris (pH 8.0), 20 mM MgCl₂, 2 mM spermidine, 10 mM DTT, 4 mM NTPs, 50 ng/μL template DNA and 1 μM T7 polymerase. The transcript was purified by 10% polyacrylamide gel electrophoresis supplemented with 8 M urea, eluted from gels by 0.5 M ammonium acetate, and precipitated by ethanol. The tRNA product redissolved in a buffer consisting of 20 mM Tris (pH 8.0) and 1 mM EDTA was heated at 65 °C for 5 min and then refolded by slowly cooling to room temperature after the

addition of 10 mM MgCl$_2$. The refolded tRNA (~ 10 mg/mL) was aliquoted and stored at −80 °C for further use.

**Preparation of in vivo produced tRNA$^{\text{Ile}}$.** The synthetic *E. coil* tRNA$^{\text{Ile}}$(GAU) gene with the native base pair A1-U72 substituted with G1-C72 was inserted between the T7 promoter and terminator of pET29b by homologous recombination. The transformed *E. coli* BL21 (DE3) cells were cultured in LB medium until the OD$_{600}$ reached ~0.6, and then 1 mM IPTG was added to induce the overexpression of tRNA at 30 °C for 16 h. The tRNA transcript was extracted from cell pellets using RNAiso Plus (TakaRa) and chloroform and precipitated from aqueous fractions using isopropanol. The redissolved tRNA was loaded into a HiTrap Q XL (GE healthcare) and eluted with a linear gradient of NaCl (0.5-0.9 M) in 20 mM Tris-HCl (pH 8.0) and 10 mM MgCl$_2$. The fractions containing tRNA were concentrated and loaded for further purification by 8 M urea/10% polyacrylamide gel electrophoresis. The tRNA was identified by UV shadowing, eluted by 0.5 M sodium acetate (pH 5.5), and then precipitated by ethanol. After annealing, tRNA was concentrated to 10 mg/mL and stored at −80 °C in buffer containing 10 mM HEPES (pH 7.5) and 10 mM MgCl$_2$.

**Isothermal titration calorimetry.** The affinity of RM-A to IleRS and the impact of substrates (L-isoleucine, ATP, or tRNA$^{\text{Ile}}$) or intermediate analogue Ile-AMS on the affinity were measured by using a MicroCal VP-ITC microcalorimeter (MicroCal). The protein and ligand samples were prepared in ITC buffer consisting of 20 mM HEPES (pH 7.5), 150 mM NaCl, 10% glycerol, 5 mM β-ME, and 0.1% DMSO. RM-A (100 μM) was titrated to 10 μM full-length *Sc*IleRS at 28 °C, with 5 μL for the first injection and 10 μL for the next 24 injections. The interval between two injections was 240 s. The disassociation constants ($K_d$) were determined by fitting the calorimetric data to a one-site binding model using Origin for ITC software (version 7). To detect the binding of RM-A to IleRS in the presence of Ile-AMS, L-isoleucine, ATP or tRNA$^{\text{Ile}}$, 10 μM full-length *Sc*IleRS preincubated with 20 μM Ile-AMS, 1 mM L-isoleucine, 2 mM ATP or 15 μM tRNA$^{\text{Ile}}$ was placed in the sample cell, a mixture of 80 μM RM-A with 20 μM Ile-AMS, 1 mM L-isoleucine, 2 mM ATP or 15 μM tRNA$^{\text{Ile}}$ was placed in the syringe, and then the titration assays were performed as described above. Besides, 10 mM MgCl$_2$ was supplemented to the protein and ligand samples when measuring the impact of tRNA$^{\text{Ile}}$ on the binding of RM-A to IleRS, and tRNA$^{\text{Pro}}$ was titrated as a control.

To reduce the consumption of the proteins and ligands, the affinity of RM-A to *Sc*IleRS variants was measured by using a MicroCal PEAQ-ITC microcalorimeter (a low-volume isothermal titration calorimeter, 200 μL cell compared to 1.4 mL cell of VP-ITC). RM-A (120 or 160 μM) was titrated to full-length *Sc*IleRS or its variants (16–20 μM) at 28 °C, with 0.2 μL for the first injection and 2 μL for the next 19 injections. The interval between the two injections was 150 s. The disassociation constants ($K_d$) were determined by fitting the calorimetric data to a one-site binding model using MicroCal PEAQ-ITC analysis software.

**ATP consumption assay.** ATP consumption assay was employed to evaluate the enzymatic activities of recombinant proteins. The reactions containing 40 nM full-length *Sc*IleRS or 100 nM C-terminal truncated *Sc*IleRS, 200 μM ATP, 1 mM L-isoleucine, 1 mg/mL *E. coli* tRNA$^{\text{Ile}}$ (produced in vivo), 30 mM HEPES (pH 7.5), 150 mM NaCl, 30 mM KCl, 40 mM MgCl$_2$, 1 mM DTT and 0.1% BSA were incubated in room temperature, and 10 μL time point aliquots were mixed with 10 μL of Kinase-Glo® Max Reagent (Promega) to measure the remaining ATP. The luminescence was read on a FlexStation 3 multimode microplate reader (Molecular Devices). Time response curves (ATP consumption vs reaction time) were used to evaluate the activities of recombinant proteins.

To test the inhibition of RM-A against the aminoacylation activity of IleRS, full-length *Sc*IleRS was incubated with RM-A for 10 min at room temperature, and then substrates were added to initiate the reaction. The final 10 μL reaction mixtures containing 40 nM full-length *Sc*IleRS, 200 μM ATP, 1 mM L-isoleucine, 1 mg/mL tRNA$^{\text{Ile}}$, 30 mM HEPES (pH 7.5), 150 mM NaCl, 30 mM KCl, 40 mM MgCl$_2$, 1 mM DTT, 0.1% BSA and RM-A at different concentrations (2.5, 10, 25, 50, 100, 250, and 1000 nM). After incubation at room temperature for 5 min, 10 μL of Kinase-Glo® Max Reagent was added to each well. The luminescence intensity of the sample without inhibitor was L$_{min}$, and the luminescence intensity of the sample without IleRS was L$_{max}$. The enzyme activity of IleRS in the presence of RM-A in different concentrations was calculated as relative enzyme activity = (L$_{max}$-L)/(L$_{max}$-L$_{min}$) × 100%. The IC$_{50}$ value was calculated by fitting the curve of relative enzyme activity versus RM-A concentration using GraphPad Prism 7.

**Pre-transfer editing assay.** Eighty-microlitre reactions containing 80 nM full-length *Sc*IleRS, 6 mM L-cysteine, 250 μM ATP, 50 μg/mL PPiase, 30 mM HEPES (pH 7.5), 150 mM NaCl, 30 mM KCl, 40 mM MgCl$_2$, 1 mM DTT and RM-A at different concentrations (from 1 μM to 1.64 nM) were incubated at room temperature for 15 min. Then, 20 μL of malachite green reagent (containing 2.45 M sulfuric acid, 0.1% malachite green, 1.5% ammonium molybdate tetrahydrate, and 0.2% tween-20) was added to the reaction mixtures and incubated for 10 min, and absorbance (A) was measured at 620 nm. The absorbance of the reaction without RM-A was A$_{max}$, and the absorbance of the reaction without IleRS was A$_{min}$. The relative enzyme activity = (A-A$_{min}$)/(A$_{max}$-A$_{min}$) × 100%. The IC$_{50}$ value was

calculated by fitting the curve of relative enzyme activity versus RM-A concentration using GraphPad Prism 7.

**Molecular docking.** Molecular docking was conducted using the Molecular Operating Environment (MOE, Version 2015.10). First, the human IleRS structure was generated by the protein structure homology-modelling program of MOE using the *Sc*IleRS structure as a template. Then, the human IleRS structure was optimized, 3-D protonated and energy minimized. RM-A and Ile-AMP molecules were docked into the corresponding binding sites in human IleRS. The highest ranked binding pose was used to represent the mode of RM-A and Ile-AMP co-binding to human IleRS.

**Reporting summary.** Further information on research design is available in the Nature Research Reporting Summary linked to this article.

## Data availability

The coordinate and structural factors of the *Sc*IleRS · RM-A · Ile-AMP complex have been deposited in the Protein Data Bank under the accession code 7D5C. The class I AARS structures used for molecular replacement or structural analyses are publicly available in Protein Data Bank, including PDB IDs 1ILE, 1M83, 1H3E, 3KFL, 4AQ7, 6LPF, 1GAX, 1QQT, 1F4L, 4ARC, 4ARI and 1QU2. The source data for Fig. 2f and Supplementary Figs. 3, 4a and 8b are provided as a Source Data file. Other data are available from the corresponding authors upon reasonable request. Source data are provided with this paper.

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

## Acknowledgements

We thank the staff of BL17U1 beamline at Shanghai Synchrotron Radiation Facility (SSRF), Shanghai, China, for assistance during data collection. This research was supported by National Natural Science Foundation of China (No. 81773636), Guangdong Basic and Applied Basic Research Foundation (No. 2019A1515011571), Program for Guangdong Introducing Innovative and Entrepreneurial Teams (No. 2016ZT06Y337), Guangdong Provincial Key Laboratory of Chiral Molecule and Drug Discovery (No. 2019B030301005), and National Engineering and Technology Research Center for New drug Druggability Evaluation (Seed Program of Guangdong Province, No. 2017B090903004).

## Author contributions

B.C. and S.L. grew the crystals, solved the structure and performed the functional experiments. S.Z. and Y.J. contributed to the biochemical experiments. Q.G., J.X., and X.-L.Y. contributed to data analysis and experiment design. B.C. and H.Z. wrote the manuscript. H.Z. supervised this research. All the authors have read and approved the final version of the manuscript.

## Competing interests

The authors declare no competing interests.
