## [Peer Review File · Nature Communications]

REVIEWER COMMENTS

Reviewer #1 (Remarks to the Author):

In this paper, the authors use X-ray crystallography and biochemical analyses to provide a detailed explanation for the inhibition of eukaryotic cytoplasmic isoleucyl-tRNA synthetase (IleRS) by natural product reveromycin A (RM-A). Aminoacyl-tRNA synthetase family plays a pivotal role in protein biosynthesis and is also involved in many biological functions. The molecular mechanisms of the inhibitors of this family have important research and application value. This work solved the co-crystal structure of *S. cerevisiae* IleRS·RM-A·Ile-AMP complex to a resolution of 1.9 Å, which resolution is high enough to support fine structural analysis. RM-A cooperates with Ile-AMP for IleRS binding, and occupies the binding site for the CCA end of tRNA^{Ile}, thereby prevents the second step reaction of the isoleucylation by IleRS. Based on the information obtained from the crystal structure, ITC, TSA, mutagenesis, and enzymatic assays were performed and generated consistent results. This work is of good novelty, because it not only provides the detailed mechanism for a biologically active nature product but also reports a single site competitive inhibition mechanism at the tRNA binding site of class I synthetase for the first time.

I recommend that Nature Communications accept this article with the following minor modifications.

1. Page 3 line 66, "Cellular lives usually 65 contain a set of AARSs of at least twenty members,..." The "at least" should be removed, since some low species only contain 19 AARSs (they have no GlnRS).
2. Page 6 line 125, change "hydrogen-bonding" to "H-bonding" to unify it with other places.
3. Page 6 line 127. Compounds 5-7 not only have shorter hydrophobic chains compared with RMA, but also lack the carboxylic acid group at C1. The carboxylic acid group plays an important role, as compound 2 also shows a significant decrease in activity.
4. Page 8 line 169, change "involving" to "involved"
5. Page 20 line 490, "the strategy the same to" should be "the same strategy as".
6. Page 20 line 589, the "2" in "MgCl₂" should be subscript.
7. Page 28 line 667, remove "involving in".

Pengfei Fang, PhD

Reviewer #2 (Remarks to the Author):

In this manuscript the authors describe a quite plausible mechanism by which a natural product reveromycin A (RM-A) inhibits eukaryotic IleRS, a member of class I aaRS. In contrast to other known aaRS inhibitors, RM-A binds to the catalytic site of IleRS along with the activated Ile substrate (Ile-AMP) and likely hinders binding of tRNA^{Ile} through steric occlusion of its CCA-end. The main result is the crystal structure of the ternary complex between IleRS, Ile-AMP and RM-A determined at 1.9 Å resolution. Moreover, to probe the proposed mechanism, the authors performed a collection of experiments including activity assays (e.g. aminoacylation and in vitro translation), binding assays, thermal shift assays and mutagenesis analyses.

Overall, this is an important study that highlights a novel inhibitory mechanism for class I aaRSs and may serve as a guide how to target sites other than the catalytic groove in this important family of enzymes with drugs, which could be extremely useful in treating a variety of pathologies. The study is well designed and executed and it is worthy of publishing in Nature Communications. The findings will be of interest to those studying structure and function of aaRSs, protein synthesis and avenues for therapeutic intervention modulating these processes.

However, before it is published the authors should revise the manuscript:

1. The language and grammatical errors should be thoroughly corrected throughout.
2. I appreciate the authors provided several snapshots of the final electron density map. Nevertheless, given that so much stake is placed on the fact that IleRS is crystallized in complex with Ile-AMP and RM-A, the authors should provide a figure showing the electron density map with Fo-Fc as map coefficients and model phases obtained immediately after molecular replacement and that covers both ligands. Such map is a much stronger evidence for bound ligands than the 2Fo-Fc map provided in the paper. Alternatively, the authors could present the Fo-Fc omit map calculated after omitting both the ligand and inhibitor from the model and calculation.
3. I would suggest that RM-A is the first tRNA-site (and not tRNA site) inhibitor of class I aaRSs. It is a better, and I would argue accurate and correct, read with a hyphen. But more importantly, in that section it must be made crystal clear that the authors used homology-based structural modeling to draw conclusions about the mechanism RM-A may be employing to inhibit IleRS. The same must be done in discussion as well. In other words, after reading the current version one could think that the authors may have determined the structure of IleRS in complex with tRNA^{Ile} and then compared it to the IleRS·Ile-AMP·RM-A ternary complex.
4. "Structural basis of bacterial IleRSs resistant to RM-A" should be rephrased into something like this: "Structural basis for resistance (or insensitivity) of bacterial IleRSs to RM-A". Resistance is usually reserved for situations where an enzyme or protein that was initially sensitive to a drug, suddenly becomes resistant through mutation(s) or some other change. Besides insensitivity, other terms that could be used are unresponsiveness and indifference.
5. Wherever Kd values are reported, errors of those values must be shown as well. ITC actually provides a means of extracting those errors.
6. Supp Fig. 1 should make its way to the main text, perhaps as the final Fig. of the main text.
7. Lastly, language, language, language. Great care should be taken to correct the many errors in the text.

Reviewer #3 (Remarks to the Author):

Through high resolution crystal structure analysis of the complex between yeast IleRS, reveromycin A and IleAMP, this manuscript gives the first detailed insight into the structural basis of IleRS inhibition by RM-A. Interestingly, the RM-A is found to occupy the 3' end CCA binding site (according to modelling with a LeuRS-tRNA^{Leu} structure) and cobind with IleAMP. The authors complement the structure with biochemical, mutagenesis and biophysical data which validate the structural observations and are also consistent with previous work on RM-A derivatives. In particular they show by ITC that the kd is enhanced if the protein is pre-incubated with IleAMS and even more so with isoleucine pre-incubation. This co-operative behaviour, which is not observed with ATP, is likely to be cause isoleucine binding inducing a conformational change in the protein that favours RM-A binding. They also show that the C18 hemi-succinate extension occupied the binding site of the ATP phosphate groups, although this is not an essential interaction since the derivative RM-T, which lacks this extension, is in fact a more potent inhibitor. Possible explanations why RM-A does not inhibit bacterial IleRS are discussed.

All this information will be very useful in refining the use of RM-A and derivatives in the treatment of various diseases and avoiding resistance.

The experiments, notably the crystallography, appear to be well done.

The only question that I ask the authors to consider is whether they can explain why the compound is not active against LeuRS and ValRS.

Also, 'Cellular lives' near the bottom of page 3 should presumably be 'Living cells'

Reviewer #4 (Remarks to the Author):

The manuscript by Chen et al reports the crystal structure of *Saccharomyces cerevisiae* isoleucyl-tRNA synthetase (IleRS) bound to inhibitor reveromycin A (RM-A) and co-purified isoleucyl-AMP intermediate. The structure depicts that the RM-A binding site largely overlaps with the binding site of the CCA-end of the tRNA and further shows that RM-A, when bound to IleRS, establishes interaction with the adjacently bound Ile-AMP. Based on the structure and binding kinetics the authors concluded that RM-A competes with tRNA and ATP for binding to IleRS while Ile or Ile-AMS bound at the synthetic site promotes RM-A binding. The results are novel and interesting particularly in light of showing a new model of interaction of the AARS inhibitors. The manuscript is clear and well written.

My major concerns are:

The binding data for RM-A in the presence of tRNA are questionable, and maybe only qualitatively accurate, as the authors used in vitro transcript of the heterologous tRNA^{Ile} from *E. coli* (line 546). It is known that tRNA^{Ile} (*E. coli* and yeast) requires anticodon modifications for the activity, making in vitro transcript kinetically incompetent. The aminoacylation rates for EctRNA^{Ile} in vitro transcript by ScIleRS are missing in the manuscript (Sup Figs 4 and 5) but the reaction time scale indicates very slow rates. For measuring K_d values, in vivo produced tRNA^{Ile} should be used, preferentially SctRNA^{Ile}.

The manuscript may benefit from the detailed biophysical analysis of the active site mutants which will allow high-resolution structure-function analysis of RM-A binding. For example, are electrostatic interactions indeed less important for RM-A binding than the hydrophobic ones as it appears from Fig 2f? How it correlates with the importance of RM-A carboxylate groups for inhibition. From the presented data, the K_d values cannot be drawn as just two enzyme concentrations were compared. To correlate structure with function one would like to have the K_d values for the RM-A interaction of the mutants as was done for the WT enzyme. These data can then provide a detailed insight into the mechanism of RM-A binding (i.e. dissect the contribution of salt bridges and hydrophobic interactions to binding).

Minor comments:

Lines 104-107. Division to CP, CP1-editing domain, CP2 and CP3 is confusing. Generally, the CP part is divided into CP1, which includes the editing domain in IleRS, LeuRS and ValRS, and CP2. I would suggest a new division such as CP1 170-196 and 400-458 (I suppose 420-458 are two helices following the editing domain which generally are included into the CP1 domain); editing domain 197-399; CP2 459-526. Also, the colours depicted these fragments at Fig 1 are similar; I suggest to authors to change the colouring to visualize these fragments better.

Lines 173-176. In the case of human IleRS, even more than for ScIleRS, it appears that electrostatic interactions do not play a (highly) important role for RM-A binding. It may be relevant to support this finding with the K_d values.

Lines 578 ATP consumption assay - is it possible that the concentration of ATP was only 4 μ M? It is way below the K_m value for ATP. Why the ATP concentration is so low? The aminoacylation activity should be followed at higher ATP concentration.

Line 599 Pre-transfer editing assay – Why pre-transfer editing was tested with cysteine which is weak and biologically irrelevant substrate for IleRS editing? The activity should be tested with valine.

Response to reviewers' comments

In the point-by-point replies below, reviewers' comments are quoted in bold, responses are in regular font, and revised text passages are colored in blue.

Reviewer #1

In this paper, the authors use X-ray crystallography and biochemical analyses to provide a detailed explanation for the inhibition of eukaryotic cytoplasmic isoleucyl-tRNA synthetase (IleRS) by natural product reveromycin A (RM-A). Aminoacyl-tRNA synthetase family plays a pivotal role in protein biosynthesis and is also involved in many biological functions. The molecular mechanisms of the inhibitors of this family have important research and application value. This work solved the co-crystal structure of *S. cerevisiae* IleRS·RM-A·Ile-AMP complex to a resolution of 1.9 Å, which resolution is high enough to support fine structural analysis. RM-A cooperates with Ile-AMP for IleRS binding, and occupies the binding site for the CCA end of tRNA^{Ile}, thereby prevents the second step reaction of the isoleucylation by IleRS. Based on the information obtained from the crystal structure, ITC, TSA, mutagenesis, and enzymatic assays were performed and generated consistent results. This work is of good novelty, because it not only provides the detailed mechanism for a biologically active nature product but also reports a single site competitive inhibition mechanism at the tRNA binding site of class I synthetase for the first time. I recommend that Nature Communications accept this article with the following minor modifications.

We thank the reviewer #1 for the positive remarks on our study and recommending publication in Nature Communications after minor modifications. Please see below for our responses to the specific questions.

1. Page 3 line 66, “Cellular lives usually 65 contain a set of AARSs of at least twenty members,...”. The “at least” should be removed, since some low species only contain 19 AARSs (they have no GlnRS).

Thanks for pointing this out. The “at least” has been removed in the revised manuscript.

2. Page 6 line 125, change “hydrogen-bonding” to “H-bonding” to unify it with other places.

This suggestion is accepted.

3. Page 6 line 127. Compounds 5-7 not only have shorter hydrophobic chains compared with RMA, but also lack the carboxylic acid group at C1. The carboxylic acid group plays an important role, as compound 2 also shows a significant decrease in activity.

We agree that the carboxylic acid group at C1 also contributes to RM-A binding. We have revised the statement (page 6, line125) to “Structure-activity relationship (SAR) studies of RM-A analogues showed that derivatives with esterified C1 carboxyl group (2) or without C1-4 portion (5) only weakly inhibit the aminoacylation activity of IleRS and that further shortening of the C1-10 side chain (6, 7) will result in lower inhibition activity (Supplementary Fig. 2)^{16,17}, highlighting the important roles of these extensive hydrophobic and polar interactions between ScIleRS and the C1-10 triene acid segment for RM-A binding.”

4. Page 8 line 169, change “involving” to “involved”.

We have fixed this error in the revised manuscript.

5. Page 20 line 490, “the strategy the same to” should be “the same strategy as”.

Fixed.

6. Page 20 line 589, the “2” in “MgCl2” should be subscript.

Fixed.

7. Page 28 line 667, remove “involving in”.

Removed.

Reviewer #2

In this manuscript the authors describe a quite plausible mechanism by which a natural product reveromycin A (RM-A) inhibits eukaryotic IleRS, a member of class I aaRS. In contrast to other known aaRS inhibitors, RM-A binds to the catalytic site of IleRS along with the activated Ile substrate (Ile-AMP) and likely hinders binding of tRNA^{Ile} through steric occlusion of its CCA-end. The main result is the crystal structure of the ternary complex between IleRS, Ile-AMP and RM-A determined at 1.9 Å resolution. Moreover, to probe the proposed mechanism, the authors performed a collection of experiments including activity assays (e.g. aminoacylation and in vitro translation), binding assays, thermal shift assays and mutagenesis analyses.

Overall, this is an important study that highlights a novel inhibitory mechanism for class I aaRSs and may serve as a guide how to target sites other than the catalytic groove in this important family of enzymes with drugs, which could be extremely useful in treating a variety of pathologies. The study is well designed and executed and it is worthy of publishing in Nature Communications. The findings will be of interest to those studying structure and function of aaRSs, protein synthesis and avenues for therapeutic intervention modulating these processes.

We thank the reviewer #2 for considering our work as an important study with a potential contribution to the field of drug discovery. Our responses to specific questions are listed below.

However, before it is published the authors should revise the manuscript:

1. The language and grammatical errors should be thoroughly corrected throughout.

We have carefully checked the manuscript and employed a commercial English editing service to improve the language usage.

2. I appreciate the authors provided several snapshots of the final electron density map. Nevertheless, given that so much stake is placed on the fact that IleRS is crystallized in complex with Ile-AMP and RM-A, the authors should provide a figure showing the electron density map with $F_o - F_c$ as map coefficients and model phases obtained immediately after molecular replacement and that covers both ligands. Such map is a much stronger evidence for bound ligands than the $2F_o - F_c$ map provided in the paper. Alternatively, the authors could present the $F_o - F_c$ omit map

calculated after omitting both the ligand and inhibitor from the model and calculation.

Thanks for this suggestion. We have now updated Fig. 1 with an annealed omit electron density map of RM-A and intermediate Ile-AMP calculated with Fourier coefficients $F_o - F_c$ and contoured at 2.5σ .

C

Fig. 1 | Structure of *ScIleRS* with bound ligands.

...

c, An annealed omit electron density map of RM-A and intermediate Ile-AMP calculated with Fourier coefficients $F_o - F_c$ and contoured at 2.5σ . RM-A and Ile-AMP co-bind in the aminoacylation pocket of the catalytic domain.

3. I would suggest that RM-A is the first tRNA-site (and not tRNA site) inhibitor of class I aaRSs. It is a better, and I would argue accurate and correct, read with a hyphen. But more importantly, in that section it must be made crystal clear that the authors used homology-based structural modeling to draw conclusions about the mechanism RM-A may be employing to inhibit IleRS. The same must be done in discussion as well. In other words, after reading the current version one could think that the authors may have determined the structure of IleRS in complex with tRNA^{Ile} and then compared it to the IleRS·Ile-AMP·RM-A ternary complex.

We thank the reviewer for these suggestions for improving the accuracy of our statements. We have corrected “tRNA site inhibitor” with “tRNA-site inhibitor” in the revised manuscript.

Yes, it is homology-based structural analysis. LeuRS is one of the closest homologues of

IleRS in AARS family, and we compared the structures of IleRS·Ile-AMP·RM-A complex and LeuRS·tRNA complex to draw the conclusion that RM-A occupies the tRNA CCA end binding site of IleRS. To avoid misleading the readers, we have revised our statements in the Results (page 13, line 289) and Discussion (page 17, line 391).

“Thus, the structural comparison between the *Sc*IleRS·RM-A·Ile-AMP complex and LeuRS·tRNA suggests that by partially mimicking the binding mode of substrate tRNA, RM-A occupies the binding pocket of the tRNA 3' CCA end in the catalytic domain of IleRS, which prevents the productive binding of tRNA for aminoacylation.”

“By comparing with the structure of LeuRS bound with tRNA^{Leu}, it is suggested that rather than resembling the structure of substrate tRNA, RM-A uses a novel scaffold to partially mimic the interactions between tRNA and IleRS.”

4. "Structural basis of bacterial IleRSs resistant to RM-A" should be rephrased into something like this: "Structural basis for resistance (or insensitivity) of bacterial IleRSs to RM-A". Resistance is usually reserved for situations where an enzyme or protein that was initially sensitive to a drug, suddenly becomes resistant through mutation(s) or some other change. Besides insensitivity, other terms that could be used are unresponsiveness and indifference.

Thanks for this suggestion. The head of this section has been changed to “Structural explanation for the insensitivity of bacterial IleRS and other AARSs to RM-A”.

5. Wherever K_d values are reported, errors of those values must be shown as well. ITC actually provides a means of extracting those errors.

We have included the errors of K_d values in the revised manuscript.

6. Supp Fig. 1 should make its way to the main text, perhaps as the final Fig. of the main text.

The suggestion is taken. The schematic diagram of RM-A inhibiting eukaryotic IleRS has been move to the main text of the revised manuscript as the final figure (Fig. 6).

7. Lastly, language, language, language. Great care should be taken to correct the

many errors in the text.

Thanks for pointing this out. We have carefully checked the manuscript and employed a commercial English editing service to improve the language usage.

Reviewer #3

Through high resolution crystal structure analysis of the complex between yeast IleRS, reveromycin A and IleAMP, this manuscript gives the first detailed insight into the structural basis of IleRS inhibition by RM-A. Interestingly, the RM-A is found to occupy the 3' end CCA binding site (according to modelling with a LeuRS-tRNA^{Leu} structure) and cobind with IleAMP. The authors complement the structure with biochemical, mutagenesis and biophysical data which validate the structural observations and are also consistent with previous work on RM-A derivatives. In particular they show by ITC that the k_d is enhanced if the protein is pre-incubated with IleAMS and even more so with isoleucine pre-incubation. This co-operative behaviour, which is not observed with ATP, is likely to be cause isoleucine binding inducing a conformational change in the protein that favours RM-A binding. They also show that the C18 hemi-succinate extension occupied the binding site of the ATP phosphate groups, although this is not an essential interaction since the derivative RM-T, which lacks this extension, is in fact a more potent inhibitor. Possible explanations why RM-A does not inhibit bacterial IleRS are discussed.

All this information will be very useful in refining the use of RM-A and derivatives in the treatment of various diseases and avoiding resistance.

The experiments, notably the crystallography, appear to be well done.

We thank reviewer #3 for these positive comments. Please see below for our responses to specific questions.

The only question that I ask the authors to consider is whether they can explain why the compound is not active against LeuRS and ValRS.

Thanks for this question. Structural and sequence analyses suggest that the extensive substitution of RM-A binding residues may be in charge of the insensitivity of eukaryotic LeuRS and ValRS to RM-A. The following section has been added to the revised manuscript (page 15, line 332) and two Supplementary Figures were added.

“LeuRS and valyl-tRNA synthetase (ValRS), the closest homologues of IleRS in class I AARS, were also reported to be insensitive to RM-A⁹. We modelled RM-A to LeuRS and ValRS by overlaying the structure of the catalytic domain of the *ScIleRS*·RM-A·Ile-AMP complex to that of human LeuRS (PDB: 6LPF)²⁶ and *T. thermophilus* ValRS (PDB: 1GAX)²⁷, and many RM-A binding residues of *ScIleRS* were found to be substituted in these two AARSs (Supplementary Figs. 11a, b and 12). For example, the key hydrophobic stacking residue Phe48 of *ScIleRS* changes to Tyr in LeuRS and to Asn in ValRS, and in addition, the other key hydrophobic residue Trp529 changes to Leu in LeuRS, which could weaken the hydrophobic stacking interactions important for RM-A binding. Furthermore, the polar interactions contributed by Arg454 and Arg462 of *ScIleRS* are also partially abolished in eukaryotic LeuRS and ValRS. Moreover, some residues on α 21 of *ScIleRS* are substituted with larger residues in LeuRS and ValRS, which possibly cause clashes with the C1-10 triene acid segment of RM-A, and mutating Thr571 to either Tyr (the corresponding residue in LeuRS) or Arg (the corresponding residue in ValRS) blocked the binding of RM-A to *ScIleRS*, as shown by TSA (Supplementary Fig. 11c).”

Supplementary Figure 11 | The possible structural explanations for the insensitivity of ValRS and LeuRS to RM-A. a-b, Modelling RM-A into LeuRS (a) and ValRS (b) by superposing the structure of the catalytic domain of *ScIleRS*·RM-A·Ile-AMP complex (light blue) to that of human LeuRS (orange, PDB: 6LPF) and *T. thermophilus* ValRS (pink, PDB: 1GAX). Many RM-A binding residues of *ScIleRS* were found to be substituted to other residues in LeuRS and ValRS, which may miss some key interactions and also cause

potential clashes between LeuRS/ValRS and modelled RM-A. **c**, Thermal melting curves of wild-type, T572R and T572Y *Sc*IleRS with or without RM-A. The effect of RM-A to the T_m value of each protein was labelled. The results showed that the T572R and T572Y variants of *Sc*IleRS lost the capability for binding RM-A.

Supplementary Figure 12 | Structure-based sequence alignments of eukaryotic cytoplasmic IleRS, ValRS and LeuRS. The protein sequences of cytoplasmic IleRS, ValRS and LeuRS from *S. cerevisiae*, *C. albicans* and *H. sapiens* were aligned using Clustal Omega program³ and then manually adjusted. The secondary structures corresponding to *Sc*IleRS are displayed above the sequences.

Also, 'Cellular lives' near the bottom of page 3 should presumably be 'Living cells'

We have fixed it in the revised manuscript.

Reviewer #4

The manuscript by Chen et al reports the crystal structure of *Saccharomyces cerevisiae* isoleucyl-tRNA synthetase (IleRS) bound to inhibitor reveromycin A (RM-A) and co-purified isoleucyl-AMP intermediate. The structure depicts that the RM-A binding site largely overlaps with the binding site of the CCA-end of the tRNA and further shows that RM-A, when bound to IleRS, establishes interaction with the adjacently bound Ile-AMP. Based on the structure and binding kinetics the authors concluded that RM-A competes with tRNA and ATP for binding to IleRS while Ile or Ile-AMS bound at the synthetic site promotes RM-A binding. The results are novel and interesting particularly in light of showing a new model of interaction of the AARS inhibitors. The manuscript is clear and well written.

We thank the reviewer #4 for thinking our results are novel and interesting. Our response to specific questions is listed below.

My major concerns are:

The binding data for RM-A in the presence of tRNA are questionable, and maybe only qualitatively accurate, as the authors used in vitro transcript of the heterologous tRNA^{Ile} from *E. coli* (line 546). It is known that tRNA^{Ile} (*E. coli* and yeast) requires anticodon modifications for the activity, making in vitro transcript kinetically incompetent. The aminoacylation rates for EctRNA^{Ile} in vitro transcript by ScIleRS are missing in the manuscript (Sup Figs 4 and 5) but the reaction time scale indicates very slow rates. For measuring K_d values, in vivo produced tRNA^{Ile} should be used, preferentially SctRNA^{Ile}.

Thanks for these important comments. Our ATP consumption assay showed that in vitro transcribed tRNA^{Ile}_{GAU} is also active, although less active compared to the in vivo produced *E. coli* tRNA^{Ile}_{GAU} (see Figure A below). And we have run EMSA to confirm that the in vitro transcribed tRNA^{Ile}_{GAU} could tightly bind to ScIleRS (see Figure B below). We also measured the isotherm of titrating RM-A to ScIleRS pre-saturated with in vivo produced *E. coli* tRNA^{Ile}_{GAU} (see Figure C below). Its K_d of 926 ± 198 nM is close to the K_d ($763 \pm$

178 nM) of titrating RM-A to ScIleRS pre-saturated with in vitro transcribed *E. coil* tRNA^{Ile_{GAU}}, and is about 6-folds weaker than the K_d (164 ± 10 nM) of titrating RM-A to apo protein. We chose to present the ITC data using the in vitro transcribed *E. coil* tRNA^{Ile_{GAU}} to keep consistent with the control tRNA^{Pro} which is also transcribed in vitro. We also tried to produce *S. cerevisiae* tRNA^{Ile_{AAU}} in vivo/in vitro using the strategy similar to that for producing *E. coil* tRNA^{Ile}, but the resulted SctRNA^{Ile_{AAU}} could not be well used by ScIleRS in our ATP consumption assays, which may be due to the lack of necessary modifications of SctRNA^{Ile} overexpressed in *E. coil* or transcribed in vitro.

Figure A. ATP consumption assay of full-length ScIleRS using *E. coil* tRNA^{Ile_{GAU}} produced in vivo/in vitro as substrate.

Figure B. Electrophoretic mobility shift assay (EMSA) for evaluating the binding between in vitro transcribed tRNA^{Ile_{GAU}} and ScIleRS. 0.5 µM tRNA^{Ile} was incubated with ScIleRS in different concentrations, and then loaded to 5% native polyacrylamide gel. The gel was run at 80 V for 2 h on ice-water, and then stained by Gel-Red nucleic acid dye.

Figure C. ITC titration of RM-A to full-length ScIIeRS pre-saturated with in vivo produced *E. coli* tRNA^{Ile}_{GAU}.

Although the in vitro transcribed *E. coli* tRNA^{Ile}_{GAU} is also active in our ATP consumption assay, we did use the in vivo produced *E. coli* tRNA^{Ile} in all the ATP consumption assays (we have stated this in the Methods in the revised manuscript). The slow reaction rates shown in the previous Sup Figs. 4 and 5 were due to the low ATP concentration (4 μM) used in the experiments. We repeated these ATP consumption assays with 200 μM ATP by using Kinase-Glo Max Reagent (Promega #V6071, which can quantify the remaining ATP up to 500 μM), and the results showed that ScIIeRS could efficiently charge *E. coli* tRNA^{Ile}. We have updated Supplementary Figs. 3 and 4 with the new data.

Supplementary Figure 3 | RM-A activity.

...

b. Inhibition to the aminoacylation activity of ScIIeRS. Error bars are s.d. (n = 3).

Supplementary Figure 4 | The ATP consumption assay and thermal shift assay of C-terminal truncated *ScIleRS*. **a**, As shown in the ATP consumption assay, the C-terminal truncated *ScIleRS* (del_C, residues 1-984) lost the aminoacylation activity, while full-length *ScIleRS* (FL) is active. Error bars are s.d. (n = 3).

The manuscript may benefit from the detailed biophysical analysis of the active site mutants which will allow high-resolution structure-function analysis of RM-A binding. For example, are electrostatic interactions indeed less important for RM-A binding than the hydrophobic ones as it appears from Fig 2f? How it correlates with the importance of RM-A carboxylate groups for inhibition. From the presented data, the K_d values cannot be drawn as just two enzyme concentrations were compared. To correlate structure with function one would like to have the K_d values for the RM-A interaction of the mutants as was done for the WT enzyme. These data can then provide a detailed insight into the mechanism of RM-A binding (i.e. dissect the contribution of salt bridges and hydrophobic interactions to binding).

Thanks for these suggestions. In the revised manuscript, we have employed TSA to compare the binding of these active site mutants to RM-A with or without the substrate L-isoleucine/intermediate analogue Ile-AMS, and also used ITC to quantitatively measure the K_d of RM-A binding to these full-length *ScIleRS* mutants. To reduce the consumption of the proteins and ligands, a low-volume isothermal titration calorimeter MicroCal PEAQ-ITC microcalorimeter (200 µL cell, while VP-ITC has a 1.4 mL cell) was used. The data are provided as the Supplementary Table 2 and Supplementary Figs. 5 and 6, which revealed that both the hydrophobic and the polar interactions are important for RM-A binding. We also include some discussions about these new data.

Page 8, line 165:

“We also measured the binding of RM-A to full-length *ScIleRS* variants by employing a fluorescence-based thermal shift assay (TSA) and isothermal titration calorimetry (ITC)

(Supplementary Table 2 and Supplementary Figs. 5 and 6). These data further highlighted the importance of Trp456, Arg460, Tyr571, Trp529, Arg462, Phe48 and Pro90 for RM-A binding because the mutations on those residues reduced the affinity of RM-A to *ScIleRS* by more than 10-fold.”

Page 9, line 190:

“Mutation of these three residues, particularly Phe48 and Trp529, to Ala dramatically reduced the RM-A binding of *ScIleRS* (Fig. 2f, Supplementary Table 2 and Supplementary Figs. 5 and 6), supporting the importance of hydrophobic stacking in RM-A inhibition.”

Page 12, line 256:

“Consistently, the affinity of RM-A to the corresponding variant of full-length *ScIleRS* only slightly decreased compared with wild-type *ScIleRS* (Supplementary Fig. 6 and Supplementary Table 2). These results suggested that the interactions between the C4' carboxyl group and KMSKS loop are less important for the binding of RM-A to *ScIleRS*.”

Page 14, line 311:

“These substitutions in bacterial *IleRS*s may prevent the binding of the long triene acid segment of RM-A because the *ScIleRS* variants F191A, Y571A and T572A were shown to be less capable of binding RM-A (Fig. 2f, Supplementary Table 2 and Supplementary Figs. 5 and 6).”

Page 14, line 318:

“Consistently, TSA results showed that RM-A, both alone and together with Ile or Ile-AMS, bound to the F48Y variant of full-length *ScIleRS* weakly, and further ITC results showed that the affinity of RM-A to the F48Y variant was 17-fold lower than that to wild-type *ScIleRS* (Supplementary Table 2 and Supplementary Figs. 5 and 6).”

Supplementary Figure 5 | Thermal melting curves of ScIIERS and its variants in the presence of different ligands.

Supplementary Figure 6 | ITC titrations of RM-A to ScIIeRS and its variants.

Supplementary Table 2 | Summary of TSA and ITC results.

	ΔT_m^a ($^{\circ}$ C)					K_d
	RM-A	Ile	RM-A + Ile	Ile-AMS	RM-A + Ile-AMS	
Wild-type	7	7.9	12.7	23.4	29.5	226 ± 17 nM
W449A	8.5	7.8	14.1	24.9	30.3	343 ± 63 nM
R454A	5	6.6	10.6	21.6	26.1	499 ± 57 nM
W456A	0.1	0.7	4.9	18.8	19.2	No binding
R460A	1.3	3.1	12	25.7	27.7	14.1 ± 3.5 μ M
D527A	4.1	5.2	15.1	24.2	30.2	1.6 ± 0.2 μ M

Y571A	2.4	4	13.3	24.6	26.1	3.2 ± 0.5 μM
T572A	5.5	4.8	11.9	22.7	27.2	340 ± 45 nM
W529A	0.1	1.9	2.2	18.4	18.5	No binding
R462A	3.7	4	12.3	26	28.8	2.4 ± 0.3 μM
F48A	1.9	0.2	2.6	12.5	12.9	7.2 ± 1.3 μM
F48Y	2	4.4	9.9	21.6	24.2	3.9 ± 0.7 μM
V89A	5.7	4.7	11.9	24.7	30	238 ± 36 nM
P90A	1.6	6.4	7.1	22.1	23.2	2.3 ± 0.3 μM
F191A	7.8	8.3	14.9	26.5	32	409 ± 83 nM
W402A	5.4	6.9	12.9	24.2	30.5	473 ± 48 nM
del_KMSKS	6.1	8.5	12.3	20.2	21.1	381 ± 28 nM

^aΔT_m is the difference between the T_m values of ScIleRS variants with and without ligands.

Minor comments:

Lines 104-107. Division to CP, CP1-editing domain, CP2 and CP3 is confusing. Generally, the CP part is divided into CP1, which includes the editing domain in IleRS, LeuRS and ValRS, and CP2. I would suggest a new division such as CP1 170-196 and 400-458 (I suppose 420-458 are two helices following the editing domain which generally are included into the CP1 domain); editing domain 197-399; CP2 459-526. Also, the colours depicted these fragments at Fig 1 are similar; I suggest to authors to change the colouring to visualize these fragments better.

Thanks for these comments and suggestions. We chose to divide CP part to editing domain (CP1), CP core, CP2 and CP3 to keep consistence with many literatures regarding to the structures of class Ia AARSs. Several reports described the domain architecture of the class Ia AARSs in the similar way, such as ValRS (Fukai, S. et al. *Cell* **103**, 793-803 (2000)), IleRS (Nureki, O. et al. *Science* **280**, 578-582 (1998); Chung, S. et al. *Mol. Cells* **43**, 350-359 (2020)), and LeuRS (Liu, R. J. et al. *Nucleic Acids Res.* **48**, 4946-4959 (2020)). We agree that some colours depicted these fragments at Fig 1 are similar, and we have recoloured the CP core and CP2 subdomain to visualize better.

Fig. 1 | Structure of *ScIleRS* with bound ligands.

...

b, Overview of the *ScIleRS*·RM-A·Ile-AMP complex structure determined at a resolution of 1.9 Å. The residue numbers in the colour-coded diagram indicate domain boundaries. RM-A and Ile-AMP are shown in spherical models. The colour schemes for RM-A (green) and Ile-AMP (salmon) are the same throughout the manuscript. The α -helices forming the binding pocket of RM-A are labelled.

Lines 173-176. In the case of human IleRS, even more than for *ScIleRS*, it appears that electrostatic interactions do not play a (highly) important role for RM-A binding. It may be relevant to support this finding with the K_d values.

Thanks for this comment. Yes, the mutations of hydrophobic residues Trp455 and Trp527 affected the rescuing capability of human IleRS more dramatically than the mutations of Arg453, Arg459 and Arg461. However, the mutations destroying the electrostatic interactions still significantly disrupted the binding of human IleRS with RM-A. Especially, the mutant R459A could only rescue about 1% of protein translation inhibited

by RM-A at 3 μM . While the rescue assay uses the C-terminal truncated human IleRS, the TSA/ITC assay requires the full-length human IleRS which cannot be well expressed in *E. coli*.

Lines 578 ATP consumption assay - is it possible that the concentration of ATP was only 4 μM ? It is way below the K_m value for ATP. Why the ATP concentration is so low? The aminoacylation activity should be followed at higher ATP concentration.

Thanks for these questions. The kit we initially used can only quantify ATP up to 10 μM . Thus, we chose to perform ATP consumption assay at 4 μM ATP. In the revised manuscript, we repeated ATP consumption assay with 200 μM ATP by using Kinase-Glo[®] Max Reagent (Promega #V6071, which can quantify the remaining ATP concentration up to 500 μM). Consistent with the previous results, C-terminal truncated IleRS is still inactive at 200 μM ATP and the IC_{50} is 25.7 nM (compared to 17.3 nM when using 4 μM ATP).

The updated ATP consumption assay is described as follows.

“ATP consumption assay was employed to evaluate the enzymatic activities of recombinant proteins. The reactions containing 40 nM full-length *ScIleRS* or 100 nM C-terminal truncated *ScIleRS*, 200 μM ATP, 1 mM L-isoleucine, 1 mg/mL *E. coli* tRNA^{Ile} (produced in vivo), 30 mM HEPES pH 7.5, 150 mM NaCl, 30 mM KCl, 40 mM MgCl₂, 1 mM DTT and 0.1% BSA were incubated in room temperature, and 10 μL time point aliquots were mixed with 10 μL of Kinase-Glo[®] Max Reagent (Promega) to measure the remaining ATP. The luminescence was read on a FlexStation 3 multimode microplate reader (Molecular Devices). Time response curves (ATP consumption vs reaction time) were used to evaluate the activities of recombinant proteins.

To test the inhibition of RM-A against the aminoacylation activity of IleRS, full-length *ScIleRS* was incubated with RM-A for 10 min at room temperature, and then substrates were added to initiate the reaction. The final 10 μL reaction mixtures containing 40 nM full-length *ScIleRS*, 200 μM ATP, 1 mM L-isoleucine, 1 mg/mL tRNA^{Ile}, 30 mM HEPES (pH 7.5), 150 mM NaCl, 30 mM KCl, 40 mM MgCl₂, 1 mM DTT, 0.1% BSA and RM-A at different concentrations (2.5, 10, 25, 50, 100, 250 and 1000 nM). After incubation at room temperature for 5 min, 10 μL of Kinase-Glo[®] Max Reagent was added to each well. The luminescence intensity of the sample without inhibitor was L_{min} , and the luminescence intensity of the sample without IleRS was L_{max} . The enzyme activity of IleRS in the presence of RM-A in different concentrations was calculated as relative enzyme activity = $(L_{\text{max}} - L)/(L_{\text{max}} - L_{\text{min}}) \times 100\%$. The IC_{50} value was calculated by fitting the curve of relative enzyme activity versus RM-A concentration using GraphPad Prism 7.”

Line 599 Pre-transfer editing assay – Why pre-transfer editing was tested with cysteine which is weak and biologically irrelevant substrate for IleRS editing? The activity should be tested with valine.

Thanks for these comments. We did try both valine and cysteine in the pre-transfer editing assay. When valine was used as the substrate in our pretransfer assay (without tRNA) for 30 min at room temperature, the final absorbance at 620 nm did not significantly increase (see table below), indicating the weak hydrolysis of the uncorrected intermediate valyl-adenylate by IleRS. Our data are consistent with the previous study by Jakubowski H. and Fersht A. R. which found that although IleRS could more efficiently misactivate valine, it weakly edited the misactivated valine in the absence of tRNA. In contrast, IleRS can efficiently hydrolyze the uncorrected intermediate product cysteinyl-adenylate in the absence of tRNA (Jakubowski H., Fersht A. R. *Nucleic Acids Res.*1981, **9**, 3105-3117). Thus, cysteine was finally used in our pre-transfer editing assay.

Amino acid	IleRS (nM)	Absorbance at 620nm	ΔA_{620nm}
Cys	0	0.285	-
	80	0.442	0.157
Val	0	0.318	-
	80	0.312	-0.006
	200	0.332	0.014

REVIEWERS' COMMENTS

Reviewer #1 (Remarks to the Author):

The authors have addressed my concerns. I would recommend this revised version to be published in Nature Communications.

Reviewer #2 (Remarks to the Author):

I am satisfied with corrections and thus suggest this paper to be published.

Reviewer #3 (Remarks to the Author):

The authors have satisfactorily responded to the referees comments including doing considerable extra experimental work. I am happy with the revised and improved manuscript and think it should now be accepted.

Reviewer #4 (Remarks to the Author):

The authors have addressed my concerns.